# SALSA-RL: Stability Analysis in the Latent Space of Actions for Reinforcement Learning

## Abstract

Modern deep reinforcement learning (DRL) methods have made significant advances in handling continuous action spaces. However, real-world control systems–especially those requiring precise and reliable performance–often demand interpretability in the sense of a-priori assessments of agent behavior to identify safe or failure-prone interactions with environments. To address this limitation, we propose SALSA-RL (Stability Analysis in the Latent Space of Actions), a novel RL framework that models control actions as dynamic, time-dependent variables evolving within a latent space. By employing a pre-trained encoder-decoder and a state-dependent linear system, our approach enables interpretability through local stability analysis, where instantaneous growth in action-norms can be predicted before their execution. We demonstrate that SALSA-RL can be deployed in a non-invasive manner for assessing the local stability of actions from pretrained RL agents without compromising on performance across diverse benchmark environments. By enabling a more interpretable analysis of action generation, SALSA-RL provides a powerful tool for advancing the design, analysis, and theoretical understanding of RL systems.

## 1 Introduction

Reinforcement learning (RL) is a powerful framework for training agents to make sequential decisions directly from environment interactions Sutton (2018), and it has shown remarkable success in complex continuous control tasks Lillicrap (2015). Unlike discrete decision-making tasks (e.g., Chess or Go), real-world dynamical systems evolve continuously, often requiring fine-grained and highly accurate control inputs to ensure safe and robust operation. In these settings, even small errors in control can propagate quickly, leading to instability, unpredictable behavior, and potential system failures. Despite these risks, most existing DRL approaches Yu et al. (2023); Heuillet et al. (2021); Kalashnikov et al. (2018) do not provide explicit mechanisms for analyzing or ensuring stability—an omission that poses significant challenges in safety-critical domains.

In dynamical system control, actions play a central role as they directly influence state evolution by serving as the primary inputs driving changes in the state trajectory. While state trajectories reflect the physical behavior of the system, they often fail to reveal the reasoning behind the control decisions. Particularly, different policies can produce similar state trajectories while relying on fundamentally different decision patterns, making state-based observations insufficient for fully understanding control logic. In contrast, action dynamics, which describe how control inputs evolve over time in response to system feedback, offer a more concrete way to assess the structure of a control policy. By analyzing trends and variations in action sequences, critical patterns can emerge, such as smooth and consistent adjustments indicating stable regulation, abrupt shifts reflecting reactive strategies, or periodic oscillations suggesting transience even in stable regimes. In the Lunar Lander problem Brockman (2016), for instance, multiple strategies can achieve a successful landing, but the specific thrust and rotation sequences can differ greatly, indicating distinct approaches to stabilization and error correction.

Additionally, observing how actions evolve provides insight into a policy's ability to maintain stability by avoiding erratic decision patterns, adapting to disturbances with smooth control adjustments, and balancing

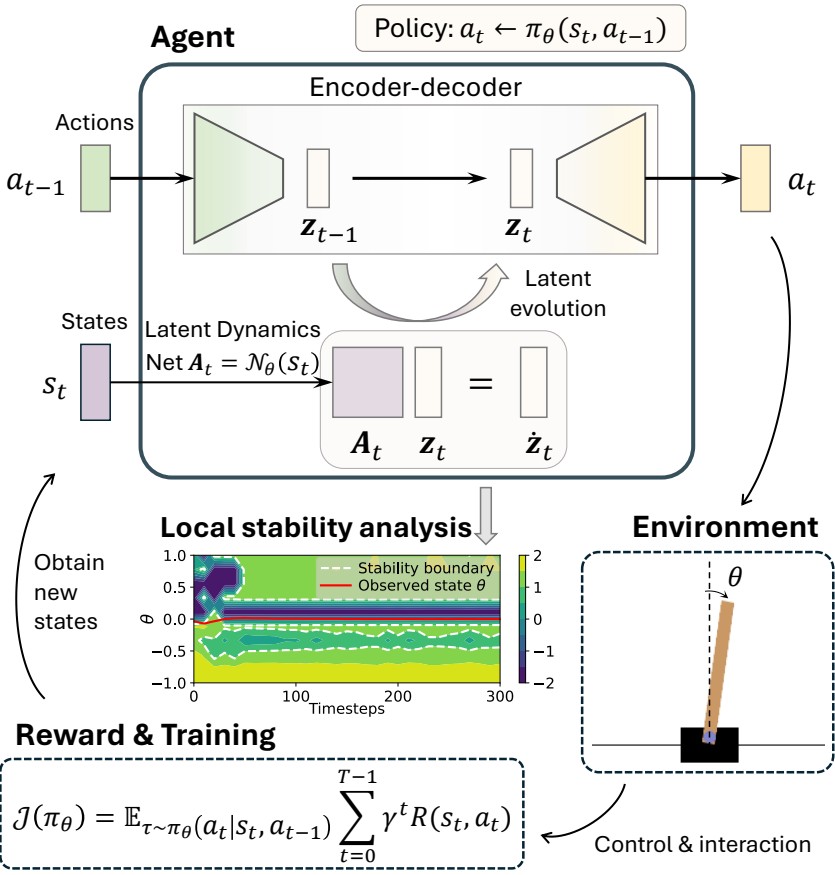

Figure 1: **Overview** of the SALSA-RL framework. Our proposed augmentation to pre-trained RL algorithms relies on a latent action representation governed by a time-varying linear dynamical system through a state-conditioned matrix $\mathbf{A}_t$. This enables local stability analyses in the action-state phase for reliable and interpretable RL deployments. The framework integrates seamlessly with existing RL algorithms, maintaining competitive performance. The contours represent the dynamically changing spectral radius of the latent linear system, with the policy seeking bounded regions (in white) with high local stability. Consequently, initializing controllers in regions outside these regions leads to high-risk behavior and potential failure.

exploration and exploitation for effective regulation without excessive correction. This analysis also enhances RL interpretability through an answer to the following questions:

1. Given pretrained reinforcement learning agents, can we identify regions of smooth or regular agent behavior in action-state phase space?
2. Conversely, can we also identify regions with a high degree of discontinuous (and consequently high-risk/failure-prone) behavior?

Identifying such patterns not only aids in policy validation but also helps detect subtle issues like overcompensation, delayed reactions, or unstable oscillations that may not be evident when observing state or action trajectories in their original high-dimensional spaces. Moreover, in many physical systems, actions are often the only fully observable or controllable signals, whereas the state may be difficult or impractical to measure directly. In addition, actions are typically subject to physical constraints or actuation budgets that may not be fully embedded during learning. Anticipating how actions behave, especially under changing conditions, can provide valuable insights for safety and planning.

In light of the importance of action dynamics for understanding control strategies, we propose SALSA-RL, a novel RL framework for dynamical system control that leverages action dynamics to analyze and design stable control strategies while enhancing interpretability. Unlike standard RL approaches, where actions are treated as discrete decisions, our framework models actions in a latent space, evolving based on the state information. We achieve this by employing a pre-trained autoencoder for action encoding and decoding. The action representation is encoded into a latent space, where the latent action dynamics $z_t$ evolve following a differential equation of the form: $\dot{z} = \mathbf{A}_t z_t$, where $\mathbf{A}_t$ is a state-dependent dynamic matrix, learned through a deep neural network conditioned on the current state. This allows for local stability analysis of action dynamics in the latent space. In particular, we define local stability in the context of a state-dependent and time-varying linear system. Stability analysis of such a system provides information about the short-term growth of action-dynamic norms. Moreover, our structured representation not only allows for local-stability analysis of control policies but also provides a means to visualize and interpret the constraints imposed on the state space, enhancing both policy transparency and generalization. Figure 1 illustrates the details of SALSA-RL.

This proposed framework is compatible with existing popular DRL methods (such as SAC, DDPG, TD3 Haarnoja et al. (2018); Fujimoto et al. (2018); Lillicrap (2015)) and can be easily integrated using exactly the same standard training strategies while maintaining competitive control performance. Beyond achieving effective control, SALSA-RL provides valuable tools for evaluating trained policies. Since actions directly influence state evolution, the influence of the action on the local stability of action dynamics serves as an indirect yet effective indicator of the stability of the state trajectories. Furthermore, as a key variable in the latent dynamics, the time-dependent square matrix $\mathbf{A}_t$ allows for the incorporation of local-in-time eigenvalue-based stability analysis. Consequently, one can perform a rigorous evaluation of the system's behavior. This analysis can provide insights into how action-space and constraints impact long-term stability and how state transitions behave near critical regions, further enhancing the interpretability of the learned control policy.

Our contributions are summarized as follows:

1. We propose SALSA-RL, a framework that models control actions as discrete-time dynamics in latent space via a time-varying linear system, enabling deeper insight into control strategies and dynamics.
2. SALSA-RL seamlessly integrates with standard DRL methods for training, maintaining performance while enhancing interpretability.
3. We perform local stability analysis to identify regions of action-state phase space for regular and high-risk behavior. We also demonstrate that pretrained agents seek out regions of regularity.

## 2 Related Work

### 2.1 Classical Control

Classical approaches for dynamical system controls include Proportional-Integral-Derivative (PID), Linear Quadratic Regulators (LQRs), and adaptive control techniquesÅström & Hägglund (2006); Swarnkar et al. (2014). These methods offer clear advantages in interpretability and stability analysis by providing explicit symbolic expressions for the relationship between states and control actions, enabling stability analysis and control law verification. However, they often struggle with high-dimensional, nonlinear, or partially observable systems where dynamics cannot be explicitly modeled Skogestad & Postlethwaite (2005).

### 2.2 Interpretability in DRL

Interpretability for machine learning algorithms has gained significant attention in recent years, particularly in healthcare, robotics, and autonomous systems Glanois et al. (2024); Yu et al. (2023); Wells & Bednarz (2021); Heuillet et al. (2021). Recent research devoted to interpretable RL has explored diverse approaches such as multi-agent systems Zabounidis et al. (2023), interpretable latent representations Chen et al. (2021), and techniques like attention mechanisms Mott et al. (2019) or genetic programming Hein et al. (2018), balancing transparency, performance, and generalizability. Below, we outline a few key directions.

**Hierarchical RL.** This approach focuses on planning and reasoning by structuring tasks into high-level (manager) and low-level (worker) policies Lyu et al. (2019); Nachum et al. (2018); Pateria et al. (2021). This modular approach excels in task decomposition, reusable subtasks, and explainable goal-setting. However, it is less suitable for specific use cases like dynamical system control, where precise and responsive low-level policies are essential Florensa et al. (2017).

**Prototype-based RL.** Prototype-based methods leverage human-defined prototypes to represent states and actions as interpretable latent features, enabling policies that balance interpretability and performance Kenny et al. (2023); Xiong et al. (2023); Yarats et al. (2021); Biehl et al. (2016). However, their reliance on manual design and lack of temporal precision limit their adaptability to dynamic environments and their ability to handle continuous, high-dimensional dynamics effectively Duan et al. (2016).

## 2.3 Dynamical System Control and Deep Reinforcement Learning

Methods such as Koopman-based control Lyu et al. (2023); Yeung et al. (2019) and Embed-to-Control Watter et al. (2015) model nonlinear dynamics through latent linearization, where the system is approximated as linear in a lifted latent space. Though not inherently RL, these methods have inspired RL frameworks with structured latent dynamics that often rely on globally fixed linear operators—rather than state-dependent ones—thereby limiting adaptability to state-dependent variations in very high-dimensional nonlinear dynamical systems Weissenbacher et al. (2022); Rozwood et al. (2024). In particular, Rozwood et al. (2024) introduces value function learning with dictionary-based methods that can provide increased insight into agent behavior. However, a direct connection to the inherent local regularity of the trained agent's behavior is not clear.

On the other hand, DRL is well-studied in physical applications, such as optimizing flow control and turbulent fluid dynamics, where domain knowledge is often available, and state-control actions exhibit stronger correlations compared to other domains Yousif et al. (2023); Weiner & Geise (2024); Garnier et al. (2021); Rabault et al. (2019). Recent approaches (summarized below) have further improved interpretability by leveraging state-action correlations and domain knowledge.

**Symbolic and Neural-Guided Approaches** Neural-guided methods like NUDGE Delfosse et al. (2024), PIRL Verma et al. (2018), and NLRL Jiang & Luo (2019) use symbolic reasoning to discover interpretable policies Jin et al. (2022). Symbolic controllers Khaled et al. (2022); Reissig & Rungger (2018), such as DSP Landajuela et al. (2021) and SINDy-RL Zolman et al. (2024), employ explicit state-control mappings to derive generalizable control laws. These methods improve interpretability and generalizability, making them suitable for safe, explainable decision-making. However, neural-guided approaches rely on logical representations Delfosse et al. (2024), limiting temporal precision and scalability in complex systems. Similarly, symbolic controllers face scalability issues in multi-dimensional or stochastic tasks where predefined forms fail to capture intricate interactions Landajuela et al. (2021). While explicit formulations may aid analysis of agent behavior, explicit stability assessments for complex high-dimensional learning tasks are challenging.

**Physics-guided RL** Physics-guided RL Liu & Wang (2021); Banerjee et al. (2023); Alam et al. (2021); Jurj et al. (2021); Cho et al. (2019); Wang et al. (2022) integrates domain knowledge and physical principles to enhance learning and performance. Approaches include informed reward functions, model-based RL with physics-based or neural network surrogate models Hernández et al. (2023); Nousiainen et al. (2021), and state design Banerjee et al. (2023). While incorporating domain knowledge can improve interpretability, these methods Alam et al. (2021); Jurj et al. (2021); Cho et al. (2019); Wang et al. (2022) often lack generalizability to other problems and require significant fine-tuning.

# 3 Method

## 3.1 Problem Formulation

Consider a control system with continuous dynamics,

$$\dot{\mathbf{s}}(t) \ = \ \mathcal{F}\big(\mathbf{s}(t), \mathbf{a}(t)\big), \tag{1}$$

where $\mathbf{s}(t) \in \mathcal{S} \subseteq \mathbb{R}^n$ is the state and $\mathbf{a}(t) \in \mathcal{A} \subseteq \mathbb{R}^m$ is the control input. Discretizing the inputs at intervals $\Delta t$ yields a discrete-time Markov Decision Process (MDP) $\mathcal{M} = (\mathcal{S}, \mathcal{A}, P, R, \gamma)$, where $\mathcal{S}$ represents the state space and $\mathcal{A}$ the action space governing the system's behavior.

Assuming the Markov property, the next state $s_{t+1}$ depends on the current state $s_t$ and action $a_t$, governed by the transition probability $P(s_{t+1} \mid s_t, a_t)$. The agent maximizes the cumulative discounted reward $R = \sum_{t=0}^{T-1} \gamma^t r(s_t, a_t)$, where $\gamma \in [0, 1]$ balances immediate and future rewards over the time horizon $T$.

Unlike standard DRL, which utilizes only state-dependent policies $\pi_\theta(\mathbf{a}_t \mid \mathbf{s}_t)$, SALSA-RL relies on actions in a latent space generated by a dynamical system, resulting in the policy $\pi_\theta(\mathbf{a}_t \mid \mathbf{s}_t, \mathbf{a}_{t-1})$. The policy is optimized to maximize the expected cumulative return:

$$J(\pi_\theta) \ = \ \mathbb{E}_{\tau \sim \pi_\theta(\mathbf{a}_t|\mathbf{s}_t, \mathbf{a}_{t-1})} \Big[ \sum_{t=0}^{T-1} \gamma^t R(\mathbf{s}_t, \mathbf{a}_t) \Big], \tag{2}$$

where $\tau = (\mathbf{s}_0, \mathbf{a}_0, \ldots, \mathbf{s}_T, \mathbf{a}_T)$ represents trajectories sampled from the policy. Although the latent action $z$ evolves in continuous time, the discrete-time MDP formulation is sufficient for the proposed algorithm, as actions are applied at discrete intervals $\Delta t$.

### 3.2 SALSA-RL Control Framework

The proposed framework simplifies control of time-dependent dynamical systems by encoding actions into a compact latent space, where state-dependent transformations govern their evolution. It consists of:

**Action Encoding and Latent Representation.** The policy's action $\mathbf{a}_t$ is encoded into a latent space using a pre-trained autoencoder:

$$\mathbf{z}_t = \text{Encoder}(\mathbf{a}_t), \tag{3}$$

where $\mathbf{z}_t \in \mathbb{R}^{d_h}$ is the latent representation of the action, $d_h$ is the latent dimension size, and $\mathbf{a}_0 = \mathbf{0}$. Here, we intentionally project actions into a higher-dimensional latent space (typically $h_d > \dim(a)$) to provide greater expressive capacity for learning structured dynamics. This higher-dimensional representation eases training and supports more interpretable stability analysis, especially in complex or underactuated tasks.

**Latent Dynamics Module.** The latent action representation evolves according to a learned state-dependent linear dynamical system $\dot{\mathbf{z}}_t = \mathbf{A}_t \mathbf{z}_t$. Simply, for discrete actions:

$$\mathbf{z}_{t+1} = \mathbf{z}_t + \mathbf{A}_t \mathbf{z}_t, \tag{4}$$
$$\mathbf{A}_t = \mathcal{N}_\theta(\mathbf{s}_t), \tag{5}$$

where $\mathbf{A}_t \in \mathbb{R}^{d_h \times d_h}$ is a state-dependent matrix learned by the neural network $\mathcal{N}_\theta$ conditioned on the current state. Importantly, to ensure numerical stability, we apply a tanh activation function followed by a scaling factor of 2 to the output of the latent dynamics network, thereby constraining all elements of $A_t$ within the range $[-2, 2]$. While this range was chosen empirically for training stability, we note that the subsequent stability analysis remains scale-invariant. We also experimented with diagonal forms of $A_t$ to simplify the analysis, but found that full (dense) matrices enabled richer inter-dependencies and better performance in control.

It is worth noting that we introduce a latent action space not for dimensionality reduction but to enable structured analysis of action behavior in the state–action phase space. This design empirically yields smoother dynamics and more stable training than operating directly in the original action space, while also allowing flexible dimensionality for constructing square dynamics matrices. It further aligns with Koopman operator theory, where systems are lifted to higher-dimensional spaces for structured modeling before being mapped back.

**Action Decoding and Control Execution.** After the linear system evolution, the latent representation is decoded back into the original action space as the actual control policy:

$$\mathbf{a}_{t+1} = \text{Decoder}(\mathbf{z}_{t+1}). \tag{6}$$

In practice, we decode $\mathbf{z}_{t+1}$ into an action $a_{t+1}$ for system control. Then, the same $a_{t+1}$ is encoded back into a new latent vector $\mathbf{z}'_{t+1}$. This mimics real-world control scenarios where only the executed action is observable, and the internal latent representation must be reconstructed based on the current state and action. This stepwise decode-encode process also reflects how a deployed agent operates under partial observability or noisy dynamics, maintaining consistency through latent transitions. Additionally, in line with standard RL environment setups, the decoded action is clipped in all our experiments. Importantly, this clipping is applied only after decoding; the latent space itself remains unconstrained.

**Policy Update and Control Loop.** The system evolves iteratively, where actions generated from the latent space update both the system state and latent variables. The framework is summarized in Algorithm 1.

---

**Algorithm 1** Latent Dynamic Control Framework

    **Input:** Initial state $\mathbf{s}_0$, initial action $\mathbf{a}_{-1} = \mathbf{0}$, time horizon $T$, pre-trained **Encoder** and **Decoder**
    **for** $t = 0$ **to** $T$ **do**
        $\mathbf{z}_t \leftarrow \mathbf{Encoder}(\mathbf{a}_{t-1})$ // Encode action
        $\mathbf{A}_t \leftarrow \mathcal{N}_\theta(\mathbf{s}_t)$
        $\mathbf{z}_{t+1} \leftarrow \mathbf{z}_t + \mathbf{A}_t \mathbf{z}_t$ // Update latent space
        $\mathbf{a}_t \leftarrow \mathbf{Decoder}(\mathbf{z}_{t+1})$ // Decode latent to next action
        $\mathbf{s}_{t+1}, Done \leftarrow$ env.step$(\mathbf{a}_t)$
        $\mathbf{a}_{t-1} \leftarrow \mathbf{a}_t$
        **if** $Done$ **then**
            **break**
        **end if**
    **end for**

---

## 3.3 Training Procedure

Training involves two stages: (1) encoder-decoder pretraining, and (2) latent dynamics optimization. **Encoder-Decoder Network.** This network is trained to create a compact and effective latent representation of actions, optimized by minimizing the reconstruction loss:

$$\mathcal{L}_{\text{recon}} = \mathbb{E}_{\mathbf{a}_t \sim D} \left[ \|\mathbf{a}_t - \hat{\mathbf{a}}_t\|^2 \right], \tag{7}$$

where $\hat{\mathbf{a}}_t$ is the reconstructed action from the decoder. To train the module, training data $D$, actions, are constructed from actions drawn either uniformly over the full action space or collected from pretrained agent. This data is then used to train the action encoding and decoding modules. We detailed this procedure in Appendix C.

**Latent Dynamic Policy.** Following this, the proposed policy is initialized in a similar manner as most DRL methods for continuous action control, such as PPO Schulman et al. (2017), SAC, DDPG, and TD3. The policy network (actor), typically generating actions in DRL as $\mathbf{a}_t = \mathcal{N}_\theta(\mathbf{s}_t)$, is instead employed to predict a state-dependent dynamic matrix $\mathbf{A}_t = \mathcal{N}_\theta(\mathbf{s}_t, \mathbf{a}_{t-1})$. The gradients of the objective to maximize the expected cumulative reward, in deterministic policies, for example, can be expressed generally as:

$$\nabla_\theta J(\pi_\theta) = \mathbb{E}_{\tau \sim \pi_\theta(\mathbf{a}_t | \mathbf{s}_t, \mathbf{a}_{t-1})} \left[ \sum_{t=0}^{T-1} \gamma^t \nabla_\theta R(\mathbf{s}_t, \mathbf{a}_t) \right] \tag{8}$$

Meanwhile, the critic network architecture, if available, remains unchanged, evaluating the state-action value function. This design allows SALSA-RL to leverage existing DRL algorithms while providing flexibility to represent and optimize latent dynamics. A detailed gradient computation is provided in Appendix A.

## 3.4 Stability Analysis and Interpretability

This section explores a range of numerical techniques for analyzing latent action dynamics, comprising the bulk of this work's contribution. This stability analysis provides insights into the system's robustness and enhances interpretability, improving understanding of the underlying agent's behavior.

**Local Stability Analysis.** While the environments are continuous-time, the action dynamics modeled in SALSA-RL follow discrete-time updates. Therefore, local stability is assessed using the spectral radius of the state-dependent linear term that evolves agent actions in the latent space. The stability analysis module in SALSA-RL is applied after training and does not interfere with the training process itself. Thus, it serves as a post-hoc diagnostic tool for pretrained policies, rather than modifying the optimization pipeline.

While formal global stability of linear time-varying systems depends on the joint spectral radius (JSR) of the operator sequence $\{I + \mathbf{A}_t\}$ Jungers (2009), computing the JSR is generally infeasible when $\mathbf{A}_t$ is state-dependent and varies continuously across time. SALSA-RL instead performs stability diagnostics locally at fixed state snapshots using the spectral radius of $\{I + \mathbf{A}_t\}$, offering a tractable and interpretable view of action evolution in specific regions of state space. Indeed, one may use this formulation to assess the spectral properties of the latent dynamics for various state-action combinations before deploying any pretrained agent.

At each time step, we compute:

$$\rho(I + \mathbf{A}_t) = \max_i |\lambda_i(I + \mathbf{A}_t)|, \tag{9}$$

Here, ***local stability*** is guaranteed when the spectral radius of this update matrix is less than one, i.e., $\rho(I + \mathbf{A}_t) < 1$. In addition to stability, we analyze the imaginary components of the eigenvalues of the matrix $A_t$, denoted $\mathrm{Im}(\lambda_i)$, to characterize oscillatory dynamics in the latent space. This helps reveal cycles or damped oscillations in control behavior, which may not be visible through magnitude alone. Specifically, local dynamics are characterized as follows:

- $\rho(I + \mathbf{A}_t) \leq 1$ with $\mathrm{Im}(\lambda_i) = 0$: non-oscillatory locally stable dynamics,
- $\rho(I + \mathbf{A}_t) \leq 1$ with $\mathrm{Im}(\lambda_i) \neq 0$: locally stable dynamics with damped oscillations,
- $\rho(I + \mathbf{A}_t) > 1$: locally unstable dynamics, regardless of the imaginary component.

Additionally, neighborhood states around the observed states $s_{\mathrm{range}}$ are evaluated by computing the corresponding $\mathbf{A}_t$ and analyzing their eigenvalues and spectral radius. This reveals locally stable or unstable regions, capturing how state variations influence stability. Importantly, our goal is not to analyze the full closed-loop environment-agent system but to empirically characterize the behavior of the learned policy through its latent action dynamics. This local stability characterization serves as a proxy for inferring how consistent, structured, or potentially unstable the policy's actions are over time, especially in high-stakes or safety-critical applications where local interpretability is crucial. This also provides insights into regions where the policy achieves stable control or displays unstable behavior. The algorithm is outlined in Appendix B.

### 3.5 Transient Growth Analysis

In the previous section, we used local stability analysis of latent (i.e., approximate) action dynamics as a proxy for assessing the local stability of a trained agent. However, even when the local stability condition $\rho(\mathbf{A}_t) < 1$ is met—non-normality (i.e., $\mathbf{A}_t\mathbf{A}_t^* \neq \mathbf{A}_t^*\mathbf{A}_t$) can still induce transient growth. This transient amplification, driven by the properties of $\mathbf{A}_t$, can destabilize intermediate states, amplify noise, or disrupt training. Due to the local nature of our stability analysis, the transient growth study becomes important for ascertaining whether a region in state-action phase space may 'escape' into a region of unstable action evolution.

For the latent dynamics described in Equation 4, the spectral radius $\rho(\mathbf{A}_t)$ is first computed to confirm the system stability. Second, for cases where the system is locally stable ($\rho < 1$) but $\mathbf{A}_t$ is non-normal, transient growth is analyzed using the Kreiss constant $\eta(\mathbf{A}_t)$, which measures the potential short-term amplification of the system. The Kreiss constant is defined as:

$$\eta(\mathbf{A}_t) = \sup_{|z|>1} \frac{|z| - 1}{\|(\mathbf{A}_t - zI)^{-1}\|}, \tag{10}$$

where $z$ is sampled from the complex plane outside the unit circle (i.e., $|z| > 1$). A high $\eta(\mathbf{A}_t)$ suggests a higher likelihood of transient growth. In such a case, one may observe an agent ultimately performing in an unstable manner even though the starting point of a trajectory has a spectral radius that is less than 1. Algorithm details are provided in Appendix B. Overall, the combination of local spectral radius computation

and Kreiss constant analysis provides a qualitative understanding of the locally transient behavior of the system, in the absence of global stability analysis. This ensures an indirect assessment of the robustness of the learned control through our proposed framework.

### 3.6 Floquet Analysis for Periodic Stability

Floquet analysis Klausmeier (2008) is a method specifically designed to evaluate the stability of periodic systems and their behavior over time. It quantifies the evolution of small perturbations in the latent space, capturing their growth, decay, or oscillatory behavior through *Floquet exponents.* This complements the spectral radius and Kreiss constant analyses by focusing on periodic stability, where transient growth alone may not capture recurring instabilities or oscillations. This analysis is applicable when the linear term in the latent dynamical system exhibits periodic trends, as seen in tasks requiring repeated oscillatory actions. An example is a pendulum failing to swing upright and instead oscillating at the bottom, where periodic stability is essential for sustained motion.

For periodic systems, where the system dynamics repeat at regular intervals such that $\mathbf{A}_{t+dt} = \mathbf{A}_t$, the state transition matrix $\Phi_t$ describes the evolution of perturbations, starting with $\Phi_0 = \mathbf{I}$ (the identity matrix). Its evolution is governed by the dynamic matrix $\mathbf{A}_t$:

$$\dot{\Phi}_t = \mathbf{A}_t \Phi_t, \tag{11}$$

$$\Phi_{t+1} = \Phi_t + \Delta t \cdot \dot{\Phi}_t. \tag{12}$$

Here, $\Delta t$ is a time discretization step, set to 1 for simplicity. At the final time horizon $T$, the state transition matrix $\Phi_T$ encodes system dynamics over one period. The eigenvalues of $\Phi_T$, known as the Floquet multipliers $\lambda_i$, are computed, and the corresponding Floquet exponents $\mu_i$ are derived as:

$$\lambda_i = \text{eigvals}(\Phi_T), \tag{13}$$

$$\mu_i = \frac{\ln(\lambda_i)}{T \cdot \Delta t}. \tag{14}$$

These exponents provide the following stability insights: $\text{Re}(\mu_i) < 0$ indicates exponential decay (stable dynamics), $\text{Re}(\mu_i) > 0$ indicates exponential growth (unstable dynamics), and $\text{Im}(\mu_i) \neq 0$ implies oscillatory behavior with a frequency proportional to $\text{Im}(\mu_i)$.

In SALSA-RL framework, $\mathbf{A}_t$ is directly derived from the state $\mathbf{s}_t$, enabling the period to be identified by analyzing state information. Floquet exponents are computed at the end of each episode to diagnose latent dynamics, revealing instability, oscillations, or unstable growth. This analysis identifies regions of instability, oscillatory behavior, or unstable growth. An intuitive application for this is for periodic systems, such as for controlling the pendulum or cartpole benchmarks in RL. A step-by-step procedure is provided in Appendix B.

## 4 Experiments

A range of classical RL environments are selected from OpenAI Gym Brockman (2016), focusing on continuous control tasks such as Cartpole-v1 (with continuous action space), Pendulum-v1, LunarLanderContinuous-v2, and BipedalWalker-v3. Additionally, a modified LunarLander is analyzed with a hovering objective to enhance stability and interpretability (details in Appendix D).

### 4.1 Eigenvalue-based Local Stability Analysis

**Pendulum.** This task involves controlling a pendulum using a single torque input at the pivot. The objective is not just to swing the pendulum upright, but to position it in the upright position and maintain balance without further oscillation or swinging. This requires precise and continuous control to counteract gravity. A latent size of $d_h = 3$ is used for model training and eigenvalue-based local stability analysis. Figure 2 shows the system's behavior over time (as a trajectory) and the eigenvalue-based evaluation (underlying contour).

In the first column of the analysis, the trajectory (red line) evolves from locally unstable regions to stable ones. Initially, as the pendulum swings up from the bottom position ($\cos\theta, \sin\theta < 0$), it exhibits instability,

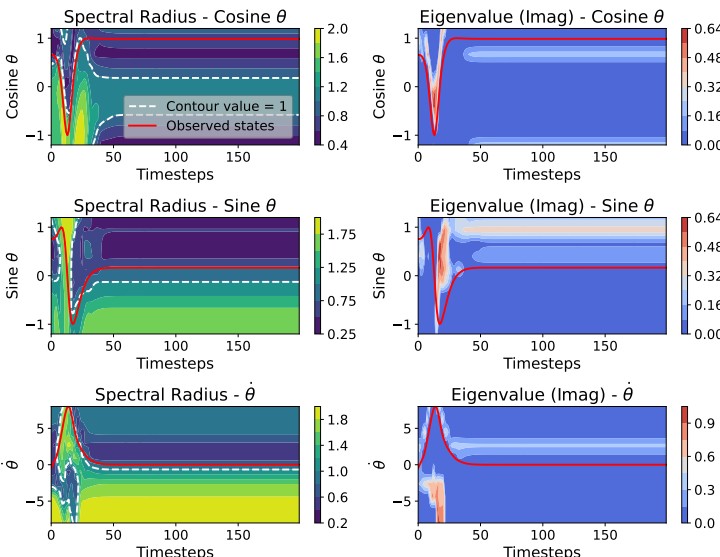

Figure 2: Local stability analysis of Pendulum control. The white line marks where the spectral radius equals 1, and the red line shows the observed state trajectory. Contours illustrate local variations in the norm behavior of latent dynamics, as captured by $\rho(\mathbf{A}_t)$. The trajectory initially enters regions of local norm growth (i.e., $\rho > 1$), but gradually shifts into regions of local contraction ($\rho < 1$), effectively maintaining the system within a range that avoids high-growth zones (bounded by the white line).

reflected by high spectral radius values ($\rho > 1$) in the action space. Over time, the pendulum stabilizes in the upright position ($\cos\theta \rightarrow 1$, $\sin\theta \rightarrow 0$), with the spectral radius dropping below 1 ($\rho < 1$), indicating entry into a stable regime and steady-state behavior. In the second column, the imaginary eigenvalues remain at 0, confirming local stability. While states initially approach regions with positive imaginary values, suggesting potential oscillations, they eventually shift away, guiding the system to a steady state.

In stabilized regimes, we observe that the latent state continues to evolve, i.e., $\mathbf{z}_{t+1} \neq \mathbf{z}_t$, despite the system being locally stable. This is due in part to the decode–re-encode cycle applied at each step: the predicted latent $\mathbf{z}_{t+1}$ is first decoded into an action $a_{t+1} = \text{Decoder}(\mathbf{z}_{t+1})$, which is then re-encoded as $\mathbf{z}'_{t+1} = \text{Encoder}(a_{t+1})$ for the next timestep. The updated latent state then evolves via Equation 4 as $\mathbf{z}_{t+2} = (I + \mathbf{A}_t)\mathbf{z}'_{t+1} \approx \mathbf{z}_{t+1}$, causing stabilized decoded actions $a_{t+2} \approx a_{t+1}$.

This phenomenon arises from the non-unique nature of the encoder–decoder mapping, along with the fact that decoding and re-encoding are performed at every step (see Section 3.2 for details on the decoder module). As a result, while the latent representation may vary, the decoded actions remain effectively invariant across time steps. This practical invariance in the action space underscores the reliability of our latent local stability diagnostics: even as latent states drift, the resulting control behavior remains consistent, highlighting the model's ability to stabilize behavior through structured latent dynamics.

Additionally, Appendix E further examines regions of local instability, recovery, and control failures, reinforcing the framework's stability analysis. Overall, this analysis shows the framework policy not only stabilizes the pendulum but does so in a manner that aligns with the primary objective of keeping it upright. By maintaining eigenvalues within the locally stable region and avoiding oscillatory dynamics, the system ensures stability while providing interpretable insights into how the policy achieves the desired control objectives.

**CartPole.** The CartPole environment involves a single discrete action that moves the cart left or right to balance the pole. With a latent dimension of $d_h = 3$, the trained model effectively learns to stabilize the pole in the upright position, as shown in Figure 3.

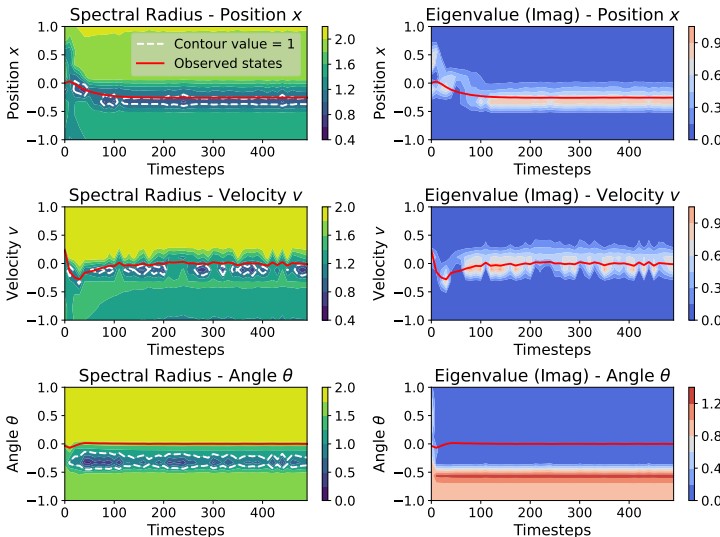

Figure 3: Local stability analysis of CartPole control. Each row shows a representative state. Due to the nature of the task, the system frequently operates in regions with $\rho(\mathbf{A}_t) > 1$, as the cart must oscillate to keep the pole upright. These excursions near or beyond the contraction boundary reflect the need for continuous corrective actions rather than convergence to a fixed point. Thus, regions with $\rho(\mathbf{A}_t) > 1$ are not necessarily undesirable, but rather reflect the non-equilibrium behavior required by the task.

However, the spectral radius tends to fluctuate near the white line $\rho = 1$, even over extended periods. This behavior can be attributed to the fact that the cart must constantly move left or right to counteract the pole's dynamics. These continual adjustments introduce small, persistent deviations in the latent dynamics, keeping the system near the edge of stability despite overall successful control. Meanwhile, the imaginary eigenvalue contours reveal a similar pattern of oscillatory behavior. The values consistently fluctuate between zero and nonzero, indicating persistent but bounded oscillations in the latent dynamics throughout the episode.

**LunarLander (Hovering Objective).** For Lunarlander experiment, we utilize a modified environment with a hovering objective instead of an objective to land safely as typically utilized. Since the state includes a 2D space, eigenvalue-based local stability is evaluated across the entire 2D domain, with each frame in Figure 4 representing a single timestep.

Among Cases 1, 3, and 4, the spectral radius consistently transitions from high-value regions to lower-value regions, reflecting a transition from locally unstable dynamics to more stable behavior over time. This gradual shift is visually indicated by the trajectory moving from red/yellow zones to cooler regions in the contour plots, especially in the later timesteps. The imaginary contour highlights regions of oscillatory behavior, and the lander's actual trajectory exhibits a transition from high to low oscillation over time. In case 2, however, the lander crashes at the final frame. The spectral radius contour in this case reveals large regions of local instability, aligning with the lander's inability to maintain its hovering status. Notably, the contour shows that the current state-action combination enters a highly unstable region, effectively signaling the risk of a crash before it occurs.

In SALSA-RL, the latent action is not clipped; the clamp is applied only after decoding. Detecting latent-space instability, therefore, pinpoints where the policy relies on saturation rather than manual clipping, as unbounded latent growth forces the decoder to flip between clipped extremes. In this modified LunarLander task, spikes in latent-action instability emerged several steps before any visible state deviation and reliably predicted impending crashes, even though the thrust values themselves remained within their clipped limits.

Overall, this strongly underscores the value of our local stability analysis, as it provides early warnings of imminent failures. Because action stability ultimately drives state evolution, latent action-space analysis

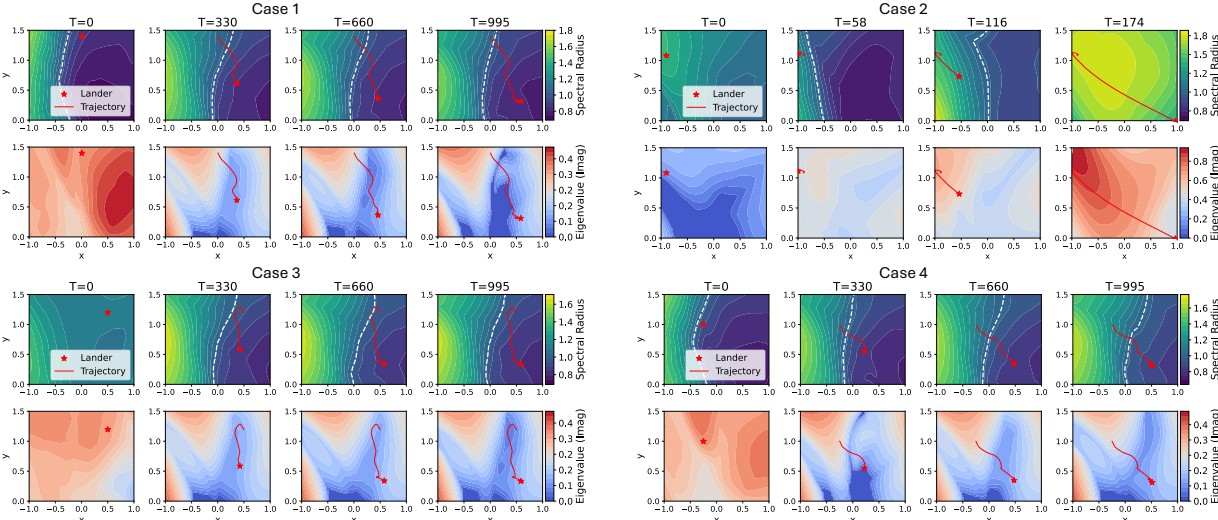

Figure 4: Local stability analysis of LunarLander hovering control. The white line indicates where spectral radius $\rho$ equals 1, separating regions of local stability and instability. The lander is initially deployed in various regions beyond the default upper-middle position (case 1). Each frame shows eigenvalue distributions in 2D space, highlighting stability regions, with the trajectory about 1,000 timesteps. Notably, case 2 contours provide early warnings of instability before the lander crashes. Animations are provided in the Supplementary.

yields policy-level interpretability, showing how the agent reacts over time rather than merely its eventual state. Empirically, in the modified LunarLander task, spikes in latent-action instability consistently preceded crashes even while the visible state trajectory still looked nominal. Overall, SALSA-RL is not a replacement for state-based analysis but a complementary, policy-aware diagnostic tool.

Besides the above 3 experiments, Appendix F explores the BipedalWalker benchmark with additional results and discussions, reinforcing the framework and analysis. Furthermore, Appendix I extends SALSA-RL to the high-dimensional Humanoid environment, where it achieves performance comparable to PPO while additionally enabling post-hoc stability analysis. These results demonstrate the scalability and robustness of the framework, highlighting how the latent-space structure supports local stability analysis and offers increasing benefits in higher-dimensional or more complex control tasks.

## 4.2 Transient Growth Analysis

Extending the local stability analysis, the Kreiss constant $\eta$ is evaluated to assess transient growth in the pendulum under a standard setup (Case 1) and a modified environment (Case 2), where gravity increases from 10 to 15 and mass from 1.0 to 1.1. The third row of Figure 5 illustrates $\eta$ over time for both cases.

In Case 1, the pendulum successfully stabilizes in the upright position. This is reflected in the actions and states, which converge to near-zero values without oscillations (first and second rows). The Kreiss constant remains small throughout, indicating minimal transient growth and robust stability.

In Case 2, the increased gravity and mass disrupt the stabilization process, causing the pendulum to exhibit periodic motion instead of reaching the upright position. This behavior is evident in the actions and states, which display sustained oscillations over time. The Kreiss constant shows periodic spikes, reflecting transient growth that aligns with the observed oscillatory response. These spikes indicate significant susceptibility to transient amplification, even though the system remains bounded over time.

This highlights the role of the Kreiss constant in capturing transient growth, particularly under non-normal dynamics. While spectral radius confirms asymptotic stability, the Kreiss constant reveals critical differences in transient behavior, providing insights into the system's robustness and susceptibility to perturbations.

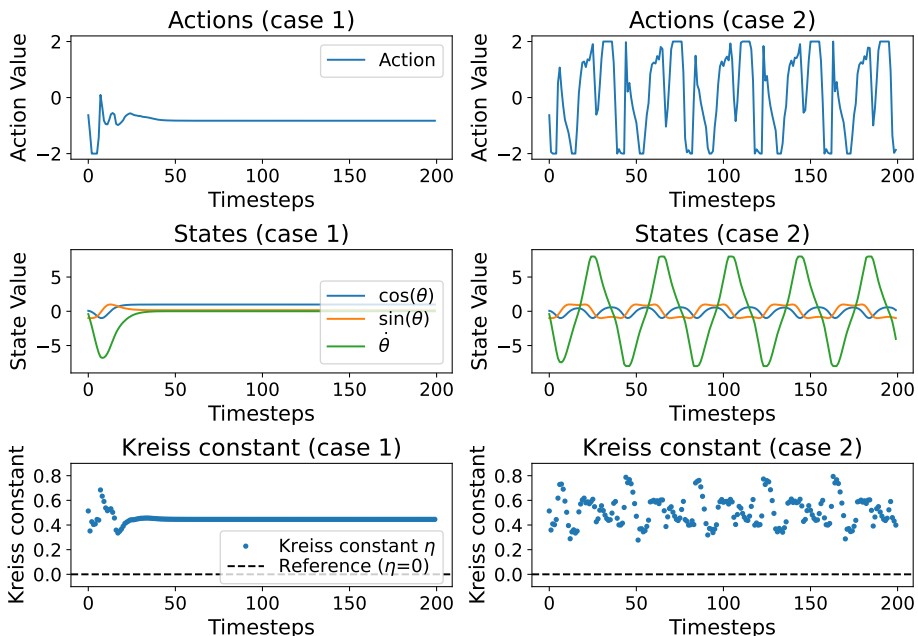

Figure 5: Transient Growth and Floquet analysis of Pendulum control. The pendulum achieves stabilization in case 1 (first column) while displaying periodic behavior with oscillations in case 2 (second column). The Kreiss constant (third row) shows differing transient growth behaviors over time. The resultant Floquet exponents are $\mu_i = 0$ and $\mu_i = \{0.96,\ 0.02 \pm 0.04i\}$ for cases 1 and 2, respectively.

### 4.3 Floquet Analysis for Periodic Stability

Building on the two pendulum control cases in Figure 5, Floquet analysis is performed to evaluate periodic stability, providing complementary insights into the stability and oscillatory behavior of the latent dynamics where metrics like eigenvalue analysis or the Kreiss constant may fall short.

In Case 1 (left column of Figure 5), the Floquet analysis reveals all exponents $\mu_i = 0$ across the three-dimensional latent space, indicating marginal stability. This means that small perturbations neither grow nor decay over the period, and the system maintains its oscillatory trajectory without exponential divergence or damping. Zero imaginary components further confirms that the observed oscillations are structural or externally driven, rather than intrinsic to the latent dynamics.

In Case 2 (right column of Figure 5, modified environment), the $\mu_i = \{0.96,\ 0.02 \pm 0.04i\}$ reveal instability, with positive real parts driving perturbation growth and imaginary components introducing oscillatory modes. These dynamics correspond to the pendulum's observed periodic motion, characterized by high-frequency control oscillations, significant angular velocity peaks, and sustained oscillatory state trajectories.

This analysis demonstrates the utility of Floquet exponents in diagnosing periodic instability and oscillatory behavior, offering deeper insight into latent action dynamics. Moreover, it potentially identifies latent space regions where policy refinement may enhance control and mitigate instability.

### 4.4 Ablation Study and Benchmark

While the goal of this work is not to produce a superior RL algorithm for specific metrics, an ablation analysis is carried out by varying the hidden dimension sizes $h_d$. The results, detailed in Table 1, demonstrate that our approach achieves interpretability while maintaining performance comparable to state-of-the-art DRL methods and symbolic approaches (DSP) Landajuela et al. (2021). Only with a severely restricted dimension

($h_d = 3$) does the method struggle with high-complexity BipedalWalker, revealing the limitations of a very low-dimensional latent space for effectively encoding the agent's action dynamics. Additional ablation studies on stability analysis are detailed in Appendix H.

Table 1: Performance comparison of SALSA-RL and various baselines. Values represent average episodic rewards over 1,000 episodes using a consistent set of environment seeds across all algorithms. Baseline results are taken from prior research Landajuela et al. (2021); Raffin (2020). The standard LunarLanderContinuous-v2 is used to ensure consistency in comparisons.

| Environment | SALSA-RL | | | | | A2C | SAC | DDPG | TD3 | PPO | DSP |
|---|---|---|---|---|---|---|---|---|---|---|---|
| | $h_d = 3$ | $h_d = 4$ | $h_d = 6$ | $h_d = 8$ | $h_d = 16$ | | | | | | |
| Pendulum | -149.2 | -149.8 | -155.8 | -157.1 | -149.9 | -162.2 | -159.3 | -169.0 | -147.1 | -154.8 | -160.5 |
| CartPole | 1000.00 | 1000.00 | 1000.00 | 1000.00 | 1000.00 | 1000.00 | 971.78 | 1000.00 | 997.98 | 993.94 | 999.59 |
| LunarLander | 257.79 | 268.25 | 246.05 | 260.82 | 242.62 | 227.08 | 272.65 | 246.24 | 225.35 | 225.12 | 251.66 |
| BipedalWalker | - | 235.11 | 280.38 | 280.91 | 262.57 | 241.01 | 307.26 | 94.21 | 310.19 | 286.20 | 264.39 |

## 5 Conclusion

This paper introduces SALSA-RL, an RL framework for local stability analysis in latent action space, leveraging action dynamics and deep learning. The utility of SALSA-RL is demonstrated across various environments, showcasing how control policies achieve stability, avoiding unstable regions, and maintaining stable states. Analysis of unstable regions reveals how key state-action combinations influence system stability, offering insights for improved control actions and reward function designs. Additional numerical analyses of periodic stability and transient growth provide a deeper understanding of underlying control outcomes, further interpreting the relationship between system states and stability.

Moreover, SALSA-RL's latent linear dynamics yield coherent action regions in phase space, reflecting consistent actions in state space (see Appendix G). Importantly, the stability analysis module functions as a post-hoc diagnostic: it is applied after training without interfering with optimization, making the approach algorithm-agnostic in principle and broadly applicable across DRL methods. SALSA-RL may also complement safe and stability-focused RL approaches Gu et al. (2024); Brunke et al. (2022); Han et al. (2020); Zhao et al. (2023); Jin & Lavaei (2020), by offering post-hoc analysis tools to interpret policy behavior without modifying the training process.

Finally, the goal of SALSA-RL is not to outperform on saturated benchmarks but to introduce an interpretable framework for empirical stability analysis of RL policies—capabilities typically lacking in existing RL algorithms. It is therefore a powerful yet simple tool for understanding the behavior of pretrained agents from various RL algorithms, offering insights into policy behavior, state transitions, and system stability. Nonetheless, SALSA-RL currently focuses on post-hoc stability analysis and does not enforce safety or correct instability during training. Extending the framework to actively guide policy updates—potentially in conjunction with safe RL methods—remains an important direction for future work.

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

## A Derivation of the Policy Gradient for SALSA-RL

### A.1 Policy Formulation

A latent dynamic control policy is defined based on Eqs. equation 3, equation 4, equation 5, and equation 6. Substituting the first three equations into the fourth yields a deterministic mapping for generating the next action $a_{t+1}$ from the current state-action pair $(s_t, a_t)$:

$$
\begin{aligned}
a_t &= \pi_\theta(s_t, a_{t-1}) \\
&= \text{Dec}(z_t) \\
&= \text{Dec}(z_{t-1} + \mathbf{A}_t z_{t-1}) \\
&= \text{Dec}\big(\text{Enc}(a_{t-1}) + \mathcal{N}_\theta(s_t)\text{Enc}(a_{t-1})\big).
\end{aligned}
\tag{15}
$$

Here, $\pi_\theta(\cdot)$ is the policy, parameterized by $\theta$, which takes the current state $s_t$ and previous action $a_t$ as input. $z_t$ represents the latent action dynamics at time $t$. $\mathcal{N}$ denotes the latent dynamics network module that takes state $s_t$ as input and outputs the time-dependent matrix $\mathbf{A}_t$.

### A.2 Policy Gradient

Using the deterministic policy gradient (DPG), we derive the gradient for the deterministic policy $\pi_\theta$, where the objective is to maximize the expected return (cumulative reward), denoted by $J(\pi_\theta)$. The gradient of this objective with respect to the neural network parameters $\theta$ can be written as:

$$
\nabla_\theta J(\pi_\theta) = \mathbb{E}_{\tau \sim \pi_\theta(\mathbf{a}_t | \mathbf{s}_t, \mathbf{a}_{t-1})}\Big[\nabla_\theta \pi_\theta(s_t, a_{t-1}) \, \nabla_a Q^\pi(s_t, a)\big|_{a=\pi_\theta(s_t, a_{t-1})}\Big],
\tag{16}
$$

where $\pi_\theta$ represents the policy parameterized by $\theta$, and $Q$ is the action-value function, which provides the expected cumulative reward starting from state $s_t$ and taking action $a_t$.

To apply Eq. equation 16 to our latent dynamic control framework, we need $\nabla_\theta \pi_\theta(s_t, a_{t-1})$.

Let us define an intermediate latent transformation as:

$$
h_\theta(s_t, a_{t-1}) = z_t = \text{Enc}(a_{t-1}) + \mathcal{N}_\theta(s_t)\text{Enc}(a_{t-1}).
\tag{17}
$$

Then

$$
\pi_\theta(s_t, a_{t-1}) = \text{Dec}(z_t) = \text{Dec}\big(h_\theta(s_t, a_{t-1})\big).
\tag{18}
$$

**Chain Rule.** From Eq. equation 16, by the chain rule,

$$
\nabla_\theta \pi_\theta(s_t, a_{t-1}) = \left.\frac{\partial \, \text{Dec}(z)}{\partial z}\right|_{z=h_\theta(s_t, a_{t-1})} \times \nabla_\theta \, h_\theta(s_t, a_{t-1}).
\tag{19}
$$

**Term 1**: $\partial \text{Dec}(z)/\partial z$. For Decoder that is fixed/pre-trained, this Jacobian is a constant. **Term 2**: $\nabla_\theta h_\theta(s_t, a_t)$. From Eq. equation 17,

$$\nabla_\theta h_\theta(s_t, a_{t-1}) = \nabla_\theta \Big[ \text{Enc}(a_{t-1}) + \mathcal{N}_\theta(s_t)\text{Enc}\big(a_{t-1}\big). \Big].$$

Assuming the Encoder is fixed, the only dependence on $\theta$ is through $\mathcal{N}_\theta(s_t)$. Hence,

$$\nabla_\theta h_\theta(s_t, a_{t-1}) = \text{Enc}(a_{t-1}) \nabla_\theta \mathcal{N}_\theta(s_t). \tag{20}$$

Putting it all together into Eq. equation 19, we obtain:

$$\nabla_\theta \pi_\theta(s_t, a_{t-1}) = \frac{\partial \text{Dec}(z)}{\partial z}\bigg|_{z=h_\theta(s_t, a_{t-1})} \times \text{Enc}(a_{t-1}) \nabla_\theta \mathcal{N}_\theta(s_t). \tag{21}$$

Substituting back into Eq. equation 16, the gradient of the objective becomes:

$$\nabla_\theta J(\pi_\theta) = \mathbb{E}_{\tau \sim \pi_\theta(\mathbf{a}_t | \mathbf{s}_t, \mathbf{a}_{t-1})}\left[ \left( \frac{\partial \text{Dec}(z)}{\partial z}\bigg|_{z=h_\theta(s_t, a_{t-1})} \times \text{Enc}(a_{t-1}) \nabla_\theta \mathcal{N}_\theta(s_t) \right) \nabla_a Q^\pi(s_t, a)\big|_{a=\pi_\theta(s_t, a_{t-1})} \right]. \tag{22}$$

This expression provides the standard DPG update rule for the latent dynamic control framework, where the policy $\pi_\theta(s_t, a_{t-1})$ is derived via the combination of encoder, latent dynamics network $\mathcal{N}_\theta$, and decoder.

For the stochastic policy, a similar derivation can be performed using the Stochastic Policy Gradient (SPG) theorem, where the gradient involves $\nabla_\theta \log \pi_\theta(a_t \mid s_t, a_{t-1})$ weighted by the action-value function $Q^\pi(s_t, a_{t-1})$.

## B Algorithms for Local Stability Analysis, Transient Growth, and Floquet Analysis

This section details additional algorithms referenced in the method section of the main article.

---

**Algorithm 2** Local Stability Evaluation

---

    **Input:** Observed $n$ states $s_t^{(i)}$, where $i = 1, 2, ..., n$
   **for** $i = 1$ **to** $n$ **do**
      **for** $t = 0$ **to** $T$ **do**
        $\hat{\mathbf{s}} \leftarrow [s_t^{(1)}, s_t^{(2)}, ..., s_t^{(n)}]$ // Construct state vector
        **for** $s$ **in** $s_{\text{range}}^{(i)}$ **do**
          $\hat{\mathbf{s}}[i] \leftarrow s$ // Replace with neighboring values
          $\mathbf{A}_t \leftarrow \mathcal{N}_\theta(\hat{\mathbf{s}})$
          $\lambda_i \leftarrow \text{eigvals}(\mathbf{A}_t)$ // Eigenvalue calculation
          $\rho(\mathbf{A}_t) \leftarrow \max_i |\lambda_i|$ // Spectral radius
          $\text{Im}_{\max} \leftarrow \max\left(\text{Im}(\lambda_i)\right)$ // Maximum imaginary magnitude
        **end for**
      **end for**
   **end for**

---

## C Encoder-Decoder Training and Discussion

The encoder and decoder each contain three layers with identical middle layer sizes. The encoder maps the action dimension to the hidden dimension $h_d$, while the decoder performs the reverse operation. Notably, we use a slightly larger number of neurons in the decoder, a common practice to enhance decoding performance and ensure a more effective reconstruction mechanism Kingma (2013).

---

**Algorithm 3** Transient Growth Evaluation

---

**Input:** Observed states $\mathbf{s}_t$ over time, error threshold $\epsilon$
**for** $t = 0$ **to** $T$ **do**
    $\mathbf{A}_t \leftarrow \mathcal{N}_\theta(\mathbf{s}_t)$
    $\lambda_1 \leftarrow \max\big(\text{eigvals}(\mathbf{A}_t)\big)$
    **if** $\lambda_1 < 1$ **then**
        normality_diff $\leftarrow \|\mathbf{A}_t \cdot \mathbf{A}_t^\top - \mathbf{A}_t^\top \cdot \mathbf{A}_t\|$
        **if** normality_diff $> \epsilon$ **then**
            $\eta(\mathbf{A}_t) \leftarrow \sup_{|z|>1} \frac{|z|-1}{\|(\mathbf{A}_t - zI)^{-1}\|}$ // Kreiss constant
        **end if**
    **end if**
**end for**

---

**Algorithm 4** Floquet Analysis for Latent Dynamics

---

**Input:** Observed states $\mathbf{s}_t$ over time, state transition matrix $\Phi \leftarrow \mathbf{I}$, identified peak-peak timesteps $t_1$ and $t_2$
**for** $t = 0$ **to** $t_2$ **do**
    $\mathbf{A}_t \leftarrow \mathcal{N}_\theta(\mathbf{s}_t)$
    **if** $t \geq t_1$ **then**
        $\Phi \leftarrow \Phi + \mathbf{A}_t \Phi$ // Update state transition matrix
    **end if**
**end for**
$\lambda_i \leftarrow \text{eigvals}(\Phi_T)$ // Spectral analysis
$\mu_i \leftarrow \frac{\ln(\lambda_i)}{T}$ // Compute Floquet exponents

---

For training, we generate action datasets from trained DLR policies, improving data efficiency compared to uniform sampling, particularly in high-dimensional settings. Training was conducted for 500 epochs with a scheduled learning rate decrease, and the final model achieved a mean squared error (MSE) of approximately $10^{-6}$ to $10^{-7}$ across different $h_d$.

In the BipedalWalker problem, which has four action dimensions, encoding and decoding with a reduced hidden dimension of $h_d = 3$ significantly increased the difficulty. Even with a properly scaled network (more neurons and layers), the training MSE loss was three orders of magnitude higher than in other cases. Since precise continuous control is crucial for tasks like BipedalWalker, we believe this partially contributed to the failure of SALSA-RL to achieve a successful control policy at $h_d = 3$ and relatively lower performance at $h_d = 4$ (see Table 1). However, this should not be viewed as a limitation of SALSA-RL, as achieving a compact network size is not our primary objective. In contrast, our stability analysis and interpretability remain consistent across different hidden dimensions, as shown in Appendix H.

## D  Modified LunarLander Environment for Hovering

In the standard LunarLanderContinuous-v2 environment, the reward function encourages efficient and accurate landings. The reward is shaped to penalize excessive fuel consumption, provide bonuses for leg contact with the ground, and reward proximity to the flat landing area. Additionally, a large bonus (+100) is given for a successful landing, while a penalty (-100) is applied for crashes or going out of bounds. This design promotes controlled and fuel-efficient landings by balancing penalties and rewards.

In the modified environment, the focus shifts from landing to hovering consistently. The bonuses for leg contacts and successful landing are removed, and a large negative penalty of -300 is applied for crashes, going out of bounds, or touching the ground. A shaping term adjusts the vertical coordinate by 0.2 to encourage hovering slightly above the surface: shaping $= -100\sqrt{x^2 + (y - 0.2)^2}$. A survival bonus of +0.2 per timestep is introduced to reward prolonged hovering, while penalties for fuel consumption remain unchanged to minimize unnecessary actions. These modifications prevent premature convergence to local reward maxima

dominated by landing or crashing. Additionally, the larger crash penalty strongly discourages failures, promoting stability and long-term control instead.

## E   Extended Stability Analysis and Interpretability on Pendulum Problem

In the main article, the analysis of pendulum control highlighted regions of local stability along with the control actions over time. In Figure 6, we examine extreme scenarios involving the absence of control actions as potential control failures, aiming to showcase extended stability analysis during these failure processes.

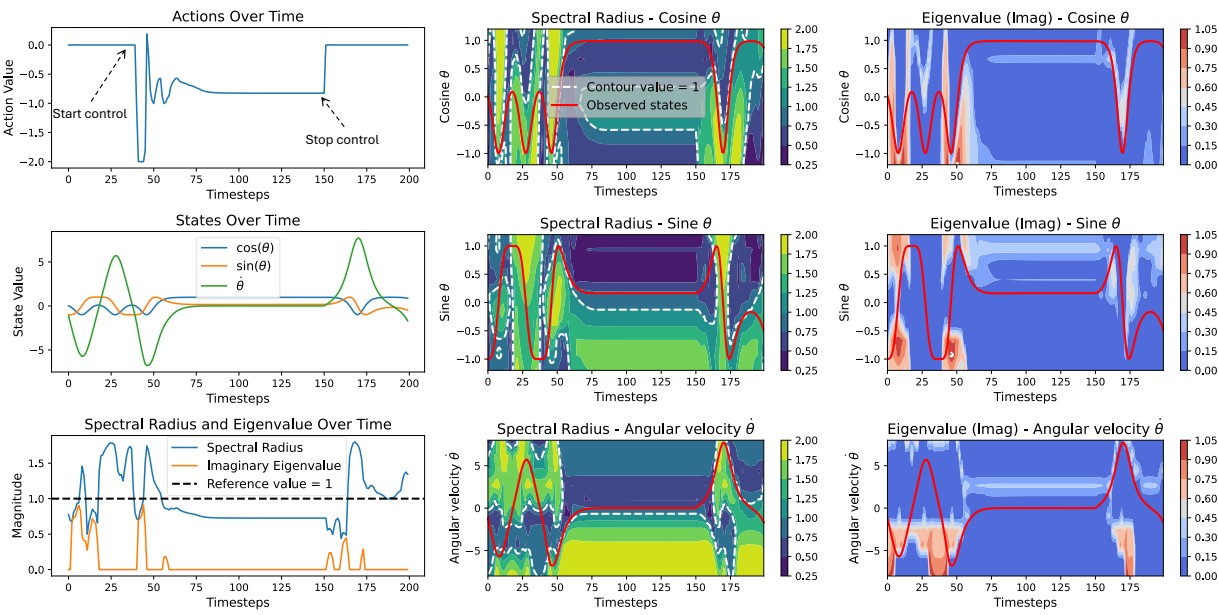

Figure 6: A custom control scenario where the system remains action-free during the first 30 and last 50 timesteps. The first column shows the evolution of action, state, and corresponding spectral radius and imaginary part of the eigenvalues, while the second and third columns present an extended stability analysis through contour visualizations.

We randomly initialized the environment and restricted actions from timestep 30 to 150. This setup enables the analysis of three typical time regions:

1. **Region of Instability (T: 0–30):** Initial local instability before any actions are applied.

2. **Region of Recovery (T: 30–150):** Control actions are applied to recover the system from local instability to stability.

3. **Region of Control Failure (T: 150–200):** The control is removed, and the system transitions back from local stability to instability.

From the spectral radius and eigenvalue plot (first column, third row), regions of local instability ($\rho_1 > 1$) are evident in time Regions 1 and 3, as well as at the initial stage of Region 2. In Region 2, the policy successfully recovers the system from local instability to stability, aligning with the above-defined phases and the pendulum control objective. For the imaginary part of the eigenvalue evolution, pronounced oscillatory behavior is observed in Regions 1 and 3, which aligns with the corresponding periods of local instability identified by the spectral radius analysis.

In the second and third columns of the figure, contour plots further validate this observation. Regions of local instability or oscillation near the trajectory are visibly highlighted in yellow and red. These appear on

both the left and right sides of the contour plots, corresponding to time regions 1 and 3, and reinforce our earlier local stability analysis and definition. Overall, this analysis significantly enhances our understanding of control failures, showcasing the interpretability and robustness of the proposed framework.

## F   Local Stability Analysis for the BipedalWalker Control

The Bipedal Walker problem involves numerous states and four actions controlling the legs and joints, aiming to achieve stable, energy-efficient forward locomotion across uneven terrain without falling. Similar to other environments analyzed in the main article, we present the local stability analysis for two different cases, as shown in Figure 7. Due to the large number of states, we illustrate only four representative states associated with the walker's hull.

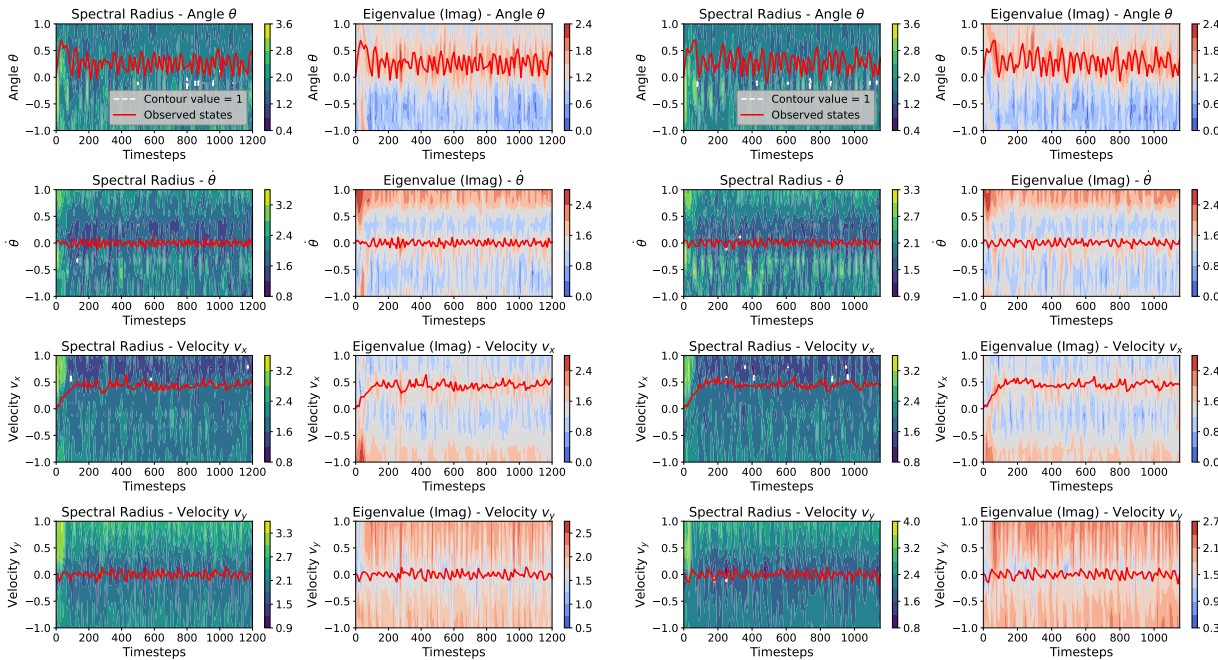

Figure 7: Local stability analysis of the Bipedal Walker control for two cases (left two columns and right two columns). Representative states associated with the walker's hull are shown. Both cases exhibit similar control patterns and local stability. The observed states (red lines) indicate that the hull maintains a positive angle, while angular velocity and vertical velocity (second row) oscillate around 0. The horizontal velocity (third row) remains positive, enabling forward movement. The large variations in angle and vertical velocity align closely with the local instability observed in the spectral radius contour, which is primarily driven by strong oscillations, as indicated by the imaginary eigenvalue contour.

In both cases, the hull primarily exhibits local instability throughout most of the episode, with the spectral radius remaining significantly greater than 1. These high values are driven largely by the magnitude of the imaginary components of the eigenvalues, even though the real parts remain mostly below 1. From the walker's perspective, this instability, caused by persistent oscillations, corresponds to the agent's continuous rightward movement. The oscillatory behavior of the hull (i.e., vertical up-and-down motion) emerges naturally as the legs alternate between stance and swing phases. This is also evident in the angle and angular velocity states (shown in the first and second rows), where significant variations in the hull's orientation are observed.

Additionally, during movement, the vertical velocity of the hull (shown in the last row of Figure 7) is minimized and remains close to zero. This behavior is also reflected in the imaginary eigenvalue contour, where the red trajectory resides in a region of low imaginary components (depicted in blue). However, oscillations during movement cannot be entirely eliminated, resulting in non-zero imaginary eigenvalues and, consequently, a

system characterized by $\rho > 1$ and continuous use of actions for controlling the system. This example reflects how the spectral radius must be interpreted in the context of the state-action behavior of a system being controlled.

## G  Phase-Space Visualizations of SALSA-RL

For the pendulum problem, where the state space is summarized by two components–the angle $\theta$ and angular velocity $\dot{\theta}$, rather than $\theta$, $\cos\theta$, and $\sin\theta$–we analyze the phase-space behavior of the learned policies. By interacting the policy with the environment, we visualize the structure of actions assigned to different states.

Figure 8 compares the baseline SAC Raffin (2020) with the proposed SALSA-RL, both sharing similar architectures and training strategies. On the right, the SAC policy results in scattered action assignments across phase space, reflecting its probabilistic nature. In contrast, on the left, SALSA-RL exhibits larger, more coherent action regions, indicating more consistent action choices for given states. This suggests that SALSA-RL encodes a structured latent representation, leading to more predictable decision-making in phase space. These slight discrepancies arise from the encoder-decoder structure in SALSA-RL, which transforms raw actions into a learned latent space before applying them, thus altering the effective action dynamics. Notably, this transformation may also enhance the stability of the pretrained agent, though a full analysis of this effect is left for future work. Like Weissenbacher et al. (2022), our approach may also be used to perform principled data augmentation for improved behavior of RL agents in terms of regularity.

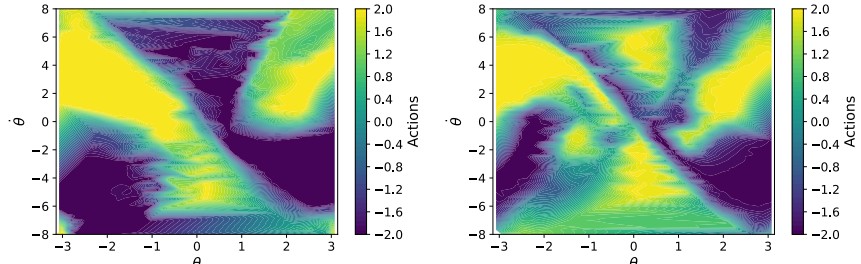

Figure 8: Action space of the proposed SALSA-RL (left) compared to the SAC baseline (right). The sharp transitions in SALSA-RL's contours suggest a more structured action space, influenced by its latent dynamic evolution, while SAC's smooth transitions reflect its probabilistic nature due to its training strategy.

To assess generalizability and robustness, we tested both policies in unseen pendulum environments by modifying internal physical parameters, such as gravity and rod mass. Compared to SAC, SALSA-RL maintained control with up to a 40% mass increase or a 20% gravity increase, while SAC failed to keep the pendulum upright in all variations. This suggests that SALSA-RL's structured latent dynamics contribute to improved stability and performance across diverse conditions.

## H  Ablation Study on Stability Analysis Across Different Hidden Dimensions

In Figure 9, we extensively show the effect of different hidden dimensions contributing to the stability analysis. Overall, the stability patterns remain consistent across different $h_d$ configurations: the trajectory initially passes through high-instability regions (yellow) before stabilizing over time.

In the first row with low $h_d$, the contour patterns are highly similar within the spectral radius and within the imaginary component plots. This consistency aligns with the main article and is particularly evident in the initial stage, where the control policy seeks local stability while crossing regions of instability. In the second row with high $h_d$, the spectral radius contour reveals more symmetric and well-defined regions of stability, in contrast to the upper row, where local stability is only observed on the lower-value side and the upper region

remains unbounded or weakly constrained. This suggests that higher latent dimensions enable more accurate local stability characterization, particularly evident in the angular velocity plot.

It is worth noting that in the higher $h_d$ cases, even after the pendulum stabilizes (beyond 50 timesteps), the imaginary component reveals regions of high values, indicating oscillatory behavior. However, these regions remain distant from the actual state of the pendulum, and the increased values are likely due to the higher-dimensional neural network introducing greater nonlinearity.

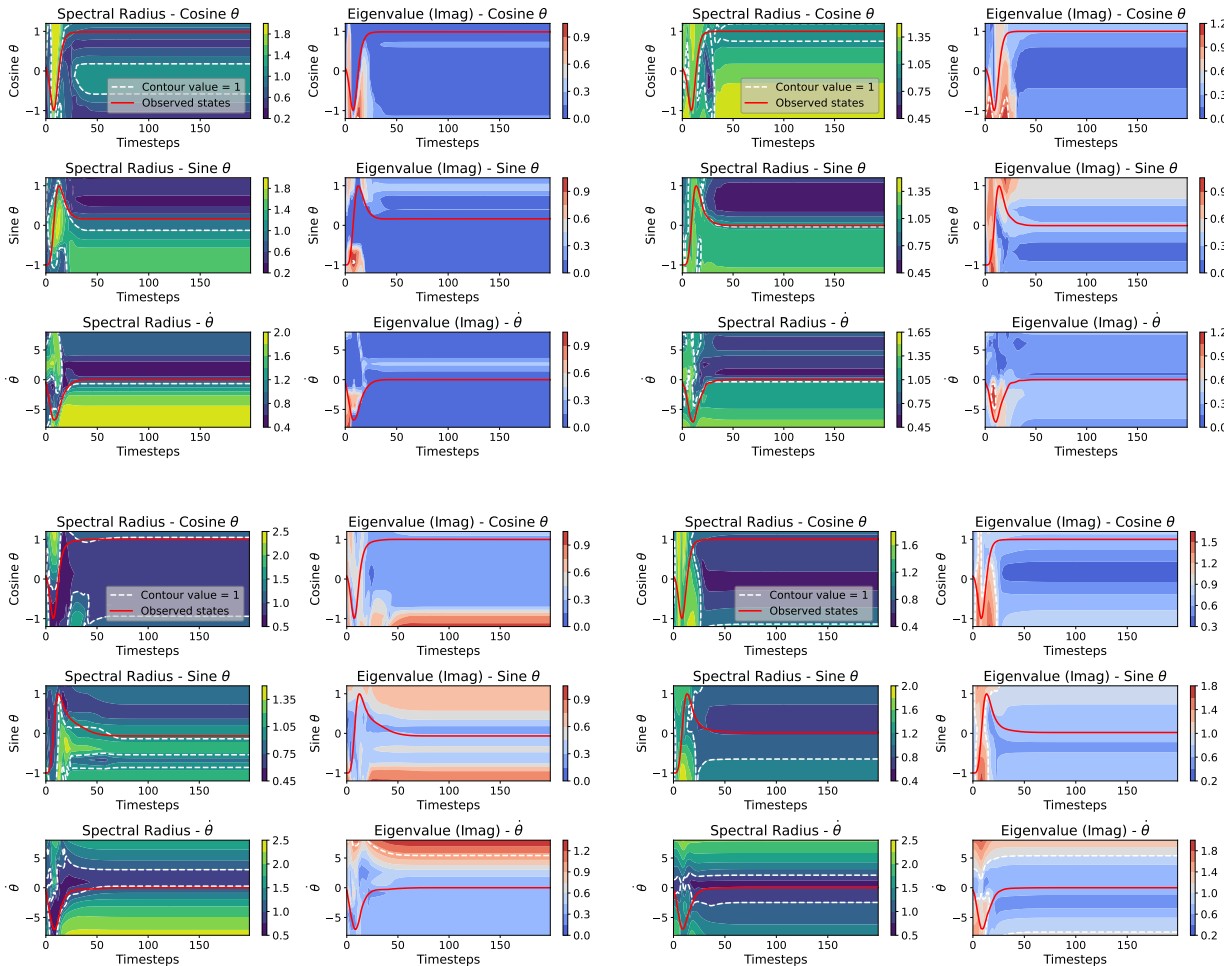

Figure 9: Local stability contour plots for different hidden dimensions: (top-left) $h_d = 3$, (top-right) $h_d = 4$, (bottom-left) $h_d = 8$, and (bottom-right) $h_d = 16$. Despite some discrepancies in numerical values, such as the real part in $h_d = 8$ showing uniform stability, similar patterns and overall behaviors hold across different configurations. This consistency reinforces the robustness of our local stability analysis, demonstrating that key trends persist regardless of the hidden dimension size.

# I  Stability Analysis on High-Dimensional Humanoid Environments

To further assess robustness, we extend SALSA-RL to high-dimensional continuous control, specifically the Humanoid task in Isaac Gym Makoviychuk et al. (2021). This environment features an action space of 21 dimensions and an observation space of 108 dimensions.

For the autoencoder, we train models with latent dimensions $h_d$ in higher values of $\{32, 64, 128\}$, since lower values significantly degrade reconstruction performance. Both the encoder and decoder are implemented as three-layer neural networks with 256 hidden units per layer.

For reinforcement learning, we adopt PPO with a policy network consisting of three layers of 400, 200, and 100 hidden units, respectively. Training is performed for 3000 epochs. Standard PPO achieves an average reward of approximately 10447 for 1000 timesteps. In comparison, SALSA-RL achieves 8999 for $h_d = 32$, 9390 for $h_d = 64$, and 9170 for $h_d = 128$.

Representatively, we show the spectral radius and imaginary eigenvalue trajectories over time for the case $h_d = 32$, as illustrated in Figure 10. The spectral radius fluctuates around the unit boundary, with frequent crossings above and below $\rho(\mathbf{A}_t) = 1$. This indicates that the learned policy operates at the edge of stability, exhibiting persistent transitions between stable and unstable regimes in the latent action dynamics. While transient excursions into instability are observed, the spectral radius tends to remain close to 1, suggesting that the overall behavior of the policy is near-critical but not dominated by instability.

In the extended analysis, we visualize the stability contours over time for the first dimension of the observation space (e.g., the first state variable), as shown in the second row of Figure 10. These plots indicate that, beyond growth rates captured by the spectral radius, imaginary components make notable contributions, reflecting oscillatory modes that influence the observed stability transitions.

Overall, the humanoid results show that SALSA-RL achieves comparable performance to PPO while additionally enabling post-hoc stability analysis in high-dimensional settings. This demonstrates the scalability and robustness of the framework, highlighting its potential for analyzing complex control policies beyond low-dimensional benchmark tasks.

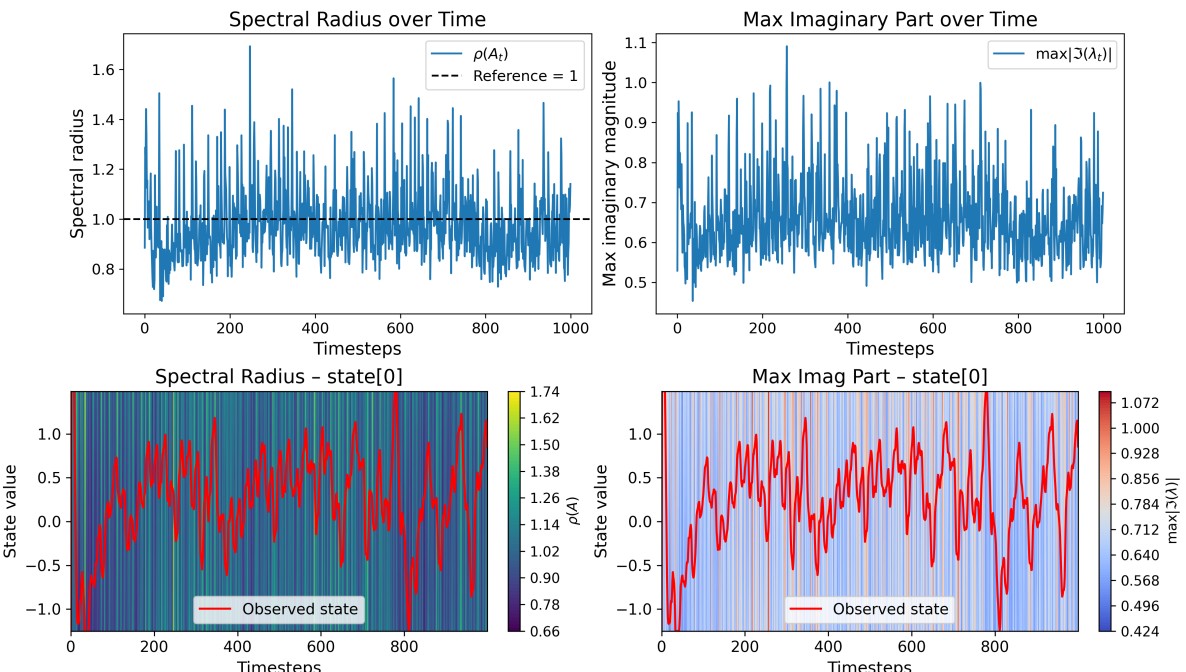

Figure 10: Stability analysis for Humanoid with $h_d = 32$. Top row: evolution of actions, states, spectral radius, and imaginary parts of eigenvalues. Bottom row: extended analysis via stability contour visualizations.

## J  SALSA-RL Complexity Analysis

For a standard two-layer actor with state dimension $n$, action dimension $m$, and hidden width $H$, the computational complexity is

$$\mathcal{C}_{\text{Actor}} = O(nH + H^2 + Hm). \tag{23}$$

For SALSA-RL, the complexity includes the latent dynamics matrix $A_t$, latent updates, and the encoder/decoder modules:

$$\mathcal{C}_{\text{SALSA-RL}} = O(nH + H^2 + Hh_d^2 + h_d^2 + 2(mc + c^2 + ch_d)), \tag{24}$$

where $h_d$ is the latent dimension and $c$ is the hidden width of the two-layer encoder/decoder.

Since $h_d \ll H \approx c$ in all our experiments (e.g., $h_d \leq 16$ while $H, c \approx 200$), the additional terms are negligible relative to the dominant $H^2$ term of the baseline. Hence, the overall complexity of SALSA-RL simplifies to

$$\mathcal{C}_{\text{SALSA-RL}} \approx O(H^2 + Hh_d^2). \tag{25}$$

