# OpenReview forum: "SALSA-RL: Stability Analysis in the Latent Space of Actions for Reinforcement Learning"
_TMLR — Rejected by TMLR_

### Review · Reviewer_ZkYN · 2025-06-16

**Summary Of Contributions:**

The authors present a method to learn time varying linear dynamic models in action latent space for RL policies that allow them to study and predict stability conditions of RL actions. They demonstrate their method in a handful of RL environments, and verify against other (vanilla) RL algorithms that their method does not damage performance in cumulative rewards.

**Audience:**

Yes

**Broader Impact Concerns:**

No concerns.

**Claims And Evidence:**

Yes

**Requested Changes:**

- I am not really sure where the motivation for analysing the stability of action (latent spaces) comes from. Usually, in classic control theoretic stability analysis, one is concerned with the stability of the state trajectories of a control system (since this is often the source of e.g. safety issues). In cyber-physical systems, states often represent variables that are more likely to be observed. Many systems are not analysable (from an observational perspective) in their actions (i.e. it is much easier to measure the angle of a robotic arm than the torque applied at a joint). Can the authors expand on this?
- In many RL problems, the action is clipped to a closed set. In this case, why is stability of the action necessary or desireable? Please correct me if I am mistaken but a locally unstable action simply implies that the action is growing in magnitude (again, locally). Is this a problem?
- The resulting linear system considered seems to be $z_{t+1}=(I+A_t)z_t $. In this case, the system would be stable if all eigenvalues of $(I+A)$ are smaller than 1 in magnitude. Authors in equation (9) only discuss the spectral radius of $A$ (not of $I+A$). Is there a reason for this?
- Some design decisions seem arbitrary. Why are the values of $A_t$ restricted to [-2,2]?
- It is not very clear to me how the action selection works in the algorithm presented. In training, is the agent applying the action $\pi(s_t)$, or the action $decoder(z_{t-1})$?

**Strengths And Weaknesses:**

## Strengths:
- The idea of learning linear latent space representations of general dynamical systems for RL applications is interesting and I believe can have many applications.
- The paper connects deep RL methods with well established control theory concepts, which is highly relevant.

## Weaknesses
- The contribution, while interesting, is thin. It is not very clear what the use of the method would be, further than gaining an (approximate) intuition of RL policies.
- Some design decisions and theory statements seem slightly arbitrary or not well justified.
- If the authors deem the main contribution to be empirical, I would expect a deeper experimental evaluation of the method (e.g. in robotic tasks).

---

> ### Author Response · Authors · 2025-08-17
>
> We would like to thank the reviewer again for the helpful feedback.
>
> As a follow-up, we have extended our evaluation to include additional experimental results on the Isaac Gym Humanoid task, presented in the appendix. These results further strengthen our claim that SALSA-RL generalizes to complex, high-dimensional control problems, even when the autoencoder is trained purely from uniformly sampled grid data rather than from a pretrained policy.

---

> > ### Comment · Reviewer_ZkYN · 2025-08-31
> >
> > Dear Authors
> >
> > I appreciate the inclusion of an extra benchmark. However, there are no references to the rest of my questions or points.

---

> > > ### Author Response · Authors · 2025-09-02
> > >
> > > We thank the reviewer again for the thoughtful feedback and follow-up. Below, we provide point-by-point references to the requested changes in the revised manuscript:
> > >
> > > **1. Motivation for action-space stability:** Addressed at the bottom of Page 2 and the bottom of Page 10 (highlighted in red).
> > >
> > > **2. Clipping and latent instability:** Discussed at the top of Page 6 and at the bottom of Page 10 (highlighted in red).
> > >
> > > **3. Stability criterion:** Already present in the original submission, on Page 7 under the "Local Stability Analysis" section.
> > >
> > > **4. Value range [-2, 2]:** Clarified on Page 6 (highlighted in red), where we explain the empirical motivation and its scale-invariance for analysis.
> > >
> > > **5. Action selection during training:** Detailed in Appendix A, Equation (15) (Page 17, unchanged from original submission).
> > >
> > > We hope these references help verify the revisions. Please let us know if further clarification would be helpful.

---

> > > > ### Comment · Reviewer_ZkYN · 2025-09-19
> > > >
> > > > Dear Authors,
> > > >
> > > > Thank you for the clarification. I do find some of the responses insufficient. In particular:
> > > >
> > > > 1. I understand that, intuitively, in a broad abstract way, instabilities are difficult to deal with, and can often lead to undesired outcomes. However, I still think that in this case the analogy with control theoretic notions of stability break down. In the case of state stability, the notion is relevant from a safety perspective because instabilities are directly linked with unbounded explosion of some state variable. Given that in classic control-theoretic problems we are interested in some form of equilibrium convergent solution to a dynamical system (e.g. we want a pendulum to be stabilised at some angle), instabilities in state are very hard to control and therefore have an impact on the controller design too. In the author's case, it is still not clear to me what an unstable latent action means for the general behaviour of an RL agent, apart from a vague notion of 'actions are increasing very fast in magnitude'. I understand that in the particular case of the LunarLander the authors link this to safety in states, but this seems cherrypicked. Sometimes this fast increase in action magnitude may be necessary? Why should actions have stable dynamics in general? (intuitively this means an action is converging to some equilibrium point).
> > > >
> > > > 2. On the stability criterion. My point is that, as the equations are formalised, the rationale seems incorrect. For a dynamical system being $z_{t+1}=A_t z_t$, the criterion for stability is that the spectral radius of A is smaller than 1. But authors write the system as $z_{t+1}=z_t + A_t z_t$. In this case, the criterion for stability is that the spectral radius of $A_t + I$ is smaller than one.

---

> > > > > ### Author Response · Authors · 2025-09-24
> > > > > **Response to Reviewer ZkYN’s comment on 19 Sept**
> > > > >
> > > > > We thank Reviewer ZkYN’s thoughtful follow-up on 19 Sept. In addition to our June 30 response, we add the following:
> > > > >
> > > > > **For comment 1:**
> > > > > We fully agree that in most RL tasks, actions are not expected to converge to a fixed point. Our framework reflects this, as seen in CartPole, where the policy makes continuous corrections near the stability boundary. Here, we use latent action stability to assess the local stability of the policy's internal dynamics, not its convergence to zero.
> > > > >
> > > > > The LunarLander case was chosen as a representative example showing how latent instability can serve as an early warning signal and reveal weaknesses in policy robustness.
> > > > >
> > > > > Also, the latent action stability correlates with more consistent decisions. As shown in Appendix G, SALSA-RL learns more coherent action regions than SAC, suggesting greater regularity and generalization when latent dynamics are stable.
> > > > >
> > > > > Regarding action magnitudes: Real action ranges are typically clipped, which can hide underlying spikes or instability. In our approach, the latent actions are unconstrained and directly observable, where clipping only occurs during decoding. This makes latent stability analysis a unique tool for diagnosing hidden instabilities in policy behavior.
> > > > >
> > > > > Lastly, instead of focusing on convergence, we emphasize the importance of long-term (or chronic) instability. This is addressed in our manuscript through Transient Growth Analysis and Floquet Analysis, which capture overshoots, oscillations, and long-term divergence beyond instantaneous behavior.
> > > > >
> > > > > **For comment 2:**
> > > > > We thank the reviewer for their careful attention to the mathematical details and for identifying this inconsistency. You are correct that our previous explanation surrounding Equation 9 was unclear and contained an error. We confirm that our implementation and all analyses are based on the correct discrete-time stability criterion, $\rho(I+\mathbf{A}_t) < 1$. The error was limited to the manuscript text and does not affect any results.
> > > > >
> > > > > We have revised the corresponding section (highlighted in red) to clarify the derivation and ensure consistency. We appreciate your feedback in helping us strengthen the mathematical clarity of the paper.

---

> ### Author Response · Authors · 2025-09-21
> **Copying Comment by Authors from Jun 30 2025**
>
> **We apologize for setting the viewing permissions wrong. The original response on Jun 30 is shown below:**
>
> We thank the reviewer for the insightful comment and for acknowledging the methodological contributions and experimental validation. Below are point-to-point responses.
>
> **Weaknesses**
>
> (1) SALSA-RL leverages the potential of linear latent representations to build another connection between deep RL and linear dynamical system analysis tools. This diagnostic capability is especially valuable in safety-critical or debugging scenarios, where understanding how a policy behaves can be as important as what it achieves, with the possibility for exploring performance in a-priori state-action combinations.
>
> (2) We appreciate the comment and have provided detailed clarifications in the manuscript.
>
> (3) We thank the reviewer for this comment. We position our method primarily as a diagnostic and interpretability tool rather than a performance-improving algorithm. To that end, we selected standard RL environments where latent instabilities and stability transitions are observable and interpretable due to long-standing study. We agree that extending the analysis to robotic or higher-DOF domains is a valuable future direction and will note this explicitly in the updated manuscript.
>
> **Requested Changes**
>
> (1) We thank the reviewer for highlighting this point. When the state is only partially observable or very high-dimensional, analysing action dynamics in a learned latent space is often more tractable and informative. Moreover, in many physical systems, actions are often the only fully observable or controllable signals, whereas the state may be difficult or impractical to measure directly. In addition, actions are typically subject to physical constraints or actuation budgets that may not be fully embedded during learning. Anticipating how actions behave, especially under changing conditions, can provide valuable insights for safety and planning.
>
> Besides, because action stability ultimately drives state evolution, latent action-space analysis yields policy-level interpretability, showing how the agent reacts over time rather than merely its eventual state. Empirically, in the modified LunarLander task, spikes in latent-action instability consistently preceded crashes even while the visible state trajectory still looked nominal. Overall, SALSA-RL is not a replacement for state-based analysis but a complementary, policy-aware diagnostic tool. We will clarify this in the introduction part of the updated manuscript.
>
> (2) We thank the reviewer for highlighting an important point. Although the decoded action is clipped in all our experiments, clipping alone does not guarantee safe or smooth control. Sudden action surges can still drive overshoot, oscillation, or divergence in the state trajectory. More importantly, they may violate the physical bounds of the actuator in a manner.
>
> In SALSA-RL, the latent action is not clipped; the clamp is applied only after decoding. Detecting latent-space instability, therefore, pinpoints where the policy relies on saturation rather than manual clipping, as unbounded latent growth forces the decoder to flip between clipped extremes. Empirically, in the modified LunarLander task, spikes in latent-action instability emerged several steps before any visible state deviation and reliably predicted impending crashes, even though the thrust values themselves remained within their clipped limits.
>
> (3) We thank the reviewer for noticing this point. Our latent dynamics stem from a forward-Euler discretisation of the continuous-time system $\dot{z}=A_t z_t$, giving $z_{t+1}=z_t+A_t z_t$. With a unit step, the discrete update matrix is $I+A_t$; local stability therefore requires $|1+\lambda_i(A_i)|<1$, which is equivalent to $\rho(A_t)<1$.
>
> We included a relevant reference in the manuscript; ensure this point is clearly explained in the updated version.
>
> [1] Jungers, Raphaël. The joint spectral radius: theory and applications. Vol. 385. Springer Science & Business Media, 2009.
>
> (4) We empirically chose to restrict $A_t$​ to [−2,2] for consistency with prior ablations, though other ranges (e.g., [−1,1]) also work well. This restriction aids stable training. However, our stability analysis, based on eigenvalue properties, is scale-invariant, and performance remains qualitatively consistent across ranges. We clarify this in the updated manuscript.
>
> (5) We thank the reviewer for the question. During training and evaluation, the agent applies the action generated by the full policy $\pi_\theta (s_t, a_{t-1})$, which incorporates the previous action through the latent dynamics formulation.
>
> As shown in Eq 15 (Appendix A.1), this corresponds to computing the latent action update and then decoding it into the final action: $a_t=Dec \big(Enc(a_{t-1})+NN_\theta(s_t)Enc(a_{t-1})\big)$. We also note that action clipping is applied immediately after decoding to ensure the action stays within valid bounds.

---

### Review · Reviewer_NH4k · 2025-06-21

**Summary Of Contributions:**

The paper introduces a reinforcement learning (RL) framework, SALSA-RL, that allows local stability analysis in the latent space of actions. SALSA-RL models continuous actions as latent dynamics by employing a pretrained encoder-decoder and a time-varying, state-dependent linear system. The proposed modeling enables interpretability and stability diagnostics of RL agents without compromising their performance. Several classical RL environments selected from OpenAI Gym are used for model evaluation.

**Audience:**

Yes

**Claims And Evidence:**

Yes

**Requested Changes:**

See **Weaknesses**.

**Strengths And Weaknesses:**

**Strengths**:
  1. The idea of performinng local stability analysis to identify regions of action-state phase space for regular and high-risk behavior seems promising.
  2. The modelling of latent action dynamics with differential equation is interesting.

**Weaknesses**:
  1. I don't understand why this framework needs latent space actions? The latent modelling is often used to reduce the dimension but the actions in most classical RL environments are already low-dimensional.
  2. On top of page 3, should it be $\dot{z}_t = A_t \times z_t$? Otherwise it would be confusing compared to Figure 1.
  3. The autoencoder is simply trained using a reconstruction loss, which would cause a large variance in the latent space modelling, i.e., small difference between original actions but large difference between latent actions. This might affect the performance.
  4. In Algorithm 1, if I understand correctly, the Encoder should be involved into the for loop as each step it will encode the action $a_t$ to $z_t$. Moreover, it might be better to just use $z_t$, $a_t$, $s_t$ rather than additional $z_{t+1}$, $a_{t+1}$, $s_{t+1}$.
  5. In experiments, since the authors added additional encoder-decoder struction for latent action modelling, it would be great to also provide the model complexity comparison.

---

> ### Author Response · Authors · 2025-09-21
> **Copying Authors’ reponses from June 30, 2025 in Response to Reviewer NH4k’s Comments**
>
> **We sincerely apologize for the earlier error in setting the viewing permissions. The original response submitted on June 30 is included below for your reference.** We appreciate your time and look forward to your comments.
>
> ----
>
> We thank the reviewer for their thoughtful comments and for recognizing the key contributions of our work, including the use of local stability analysis and latent action modeling. Please find our detailed point-by-point responses below.
>
> For **Weaknesses:**
>
> (1) We thank the reviewer for raising this point. We use a latent action space not for dimensionality reduction, but for a-priori understanding of action behaviour in state-action phase space. Furthermore, we empirically determine that this design enables smoother dynamics and more stable training, which are difficult to achieve directly in the original action space, even in standard Gym environments.
>
> The latent space also allows for flexible dimensionality, making it possible to construct square matrices for latent dynamics evolution, which is not always feasible in the original action space. This design aligns with principles in Koopman operator theory and latent dynamical systems, where systems are lifted to higher-dimensional spaces for structured modeling before being mapped back.
>
> (2) We thank the reviewer for pointing this out. You are correct that it should be $\dot{z}_t = A_t z_t$ for consistency with Figure 1. We have corrected this in the updated manuscript.
>
> (3) We thank the reviewer for highlighting this important point. Indeed, since the autoencoder is trained solely with a reconstruction loss, the encoder–decoder mapping is nonlinear and potentially non-smooth. As a result, small changes in the original action space may correspond to large variations in the latent space, potentially introducing jumps in the latent dynamics. This limitation and the non-uniqueness of the latent mapping are discussed in Section 4.1 (paragraphs 3–4).
>
> However, we find that in practice, the learned latent transitions remain sufficiently smooth and continuous for our analysis purposes, and the decoded actions, which are clipped before execution, help mitigate the effect of such jumps.
>
> (4) We thank the reviewer for this helpful observation. The encoder should be included inside the for-loop to map each action. Additionally, we noticed that $a_t$ (not $a_{t+1}$) should be derived from $z_t \leftarrow \text{Encoder}(a_t)$ and $a_t \leftarrow \text{Decoder}(z_{t+1}) $, to be consistent with Figure 1. Now, Time indexing is strictly consistent: $a_{t−1}→z_{t}→z_{t+1}→a_{t}$. We have corrected this and updated the corresponding symbols in the manuscript.
>
> (5) For standard two–layer actor: $O(nH + H^{2} + Hm)$, where $n$ and $m$ are state and action dimensions, and $H$ is the hidden width.
>
> SALSA-RL (latent matrix $A_t$, latent update, encoder/decoder): $O\bigl(nH + H^{2} + Hh_d^{2} + h_d^{2} + 2(m c + c^{2} + c h_d)\bigr),$ where $h_d$ is the latent dimension and $c$ the hidden width of the two–layer encoder/decoder.
>
> Because $h_d \ll H \approx c$ in all our experiments (e.g., $h_d \le 16$ while $H,c \approx 200$), other terms are negligible relative to the dominant $H^{2}$ term of the baseline. Hence, the overall complexity of SALSA-RL remains $O(H^{2}+H h_d^{2})$.
>
> We have included this in our updated manuscript.

---

### Review · Reviewer_6vq5 · 2025-08-03

**Summary Of Contributions:**

This paper presents SALSA-RL, a novel framework for analyzing the stability of policies in deep RL for continuous control tasks. The authors identify a critical limitation in modern RL, such as the lack of interpretability and post-hoc safety assessments, which hinders deployment in real-world dynamical systems. To address this, SALSA-RL reframes action generation. Instead of a policy directly outputting an action from a state, it models actions as variables evolving within a latent space according to a state-dependent linear dynamical system. The core idea involves using a pre-trained autoencoder to map actions to and from a latent space. Within this space, the latent action vector evolves based on a matrix $A_t$, which is predicted by a neural network conditioned on the current environment state $s_t$. This structured representation is the key to the framework's interpretability. It allows for local stability analysis of the policy's action generation process at any given state. By examining the eigenvalues of the dynamics matrix $A_t$, one can predict whether subsequent actions are likely to grow in norm (indicating potential instability), shrink (indicating stability), or exhibit oscillatory behavior. The authors demonstrate that by visualizing the spectral properties of the latent dynamics across the state space, one can identify regions of smooth, regular control versus regions prone to high-risk, discontinuous behavior. The paper extends this analysis beyond simple eigenvalue checks, incorporating methods like Kreiss constant analysis to assess transient growth in non-normal systems and Floquet analysis for evaluating the stability of periodic behaviors. An important aspect of SALSA-RL is its non-invasive nature, by showing that it can be used as a post-hoc analysis tool for pre-trained agents, providing insights into their behavior without requiring retraining.

**Audience:**

Yes

**Broader Impact Concerns:**

This paper does not have a Broader Impact Statement. Considering the theme of the paper, this may not the significant problem.

**Claims And Evidence:**

No

**Requested Changes:**

Please respond and revise the paper based on the feedback in "Weaknesses" above.
- [w1]: Critical. To clarify the scope of the paper.
- [w2]: Strengthen the paper.
- [w3]: Strengthen the paper.
- [w4]: Critical. To show the applicability to the real world tasks.
- [w5]: Critical. To clarify the relevance to the previous works and relevant literatures.

**Strengths And Weaknesses:**

### Strengths
- [s1] This paper is well-written and easy to follow.
- [s2] The stability analysis of the RL policy is beneficial and an important topic to the community toward real-world applications.
- [s3] While the main idea of the SALSA-RL (Figure 1 and Section 3.2) seems to be inspired from the control theory, the performances are comparable to DeepRL methods (Table 1), and the local stability analysis (Figure 2, 3, 4) is insightful.

### Weaknesses
- [w1] In the abstract and conclusion, the author mentioned that "a-priori assessments" and "post-hoc analysis tools to interpret policy behavior without modifying the training process." Could you clarify these parts more details? If my understanding is correct, the training procedure is different from standard RL (e.g., PPO, SAC, ...) and the paper did not demonstrate that SALSA-RL can be applied to the other algorithms. If you already explained this, please point the lines in the paper.
- [w2] Related to [w1], can we apply SALSA-RL on top of other standard RL policies, imitation learning policies, or vision-language-action models?
- [w3] SALSA-RL pre-trains Encoder and Decoder from some offline data. The details of data collection and the effects of the quality of data did not discussed in the paper. I guess the conclusion might be a bit different if we pre-train AE with expert actions from PPO and if we pre-train AE with random actions.
- [w4] The benchmark is limited to low-dimensional action space, such as Pendulum, CartPole, LunarLander, BipedalWalker. It would be great to do the same analysis on more complex, high-dimensional, and safety-critical continuous control environments, such as MuJoCo, robot arm control (e.g., Franka Kitchen, ...), Isaac Gym (quadruped, humanoid, etc), and so on. This is more important for the paper to show its value toward real-world applications. You do not need to do all the experiments, but the more the better.
- [w5] While this paper analyze the stability of the RL policy, it did not discuss or propose how to control it, or fix it if the system is unstable. Also, related to the safe RL literature, SALSA-RL was not tested on the constrained MDP. Because this paper is about the stability in deep RL, the comparison against safe RL, and how proposed methods can be useful to improve the stability would be mandatory.

---

> ### Author Response · Authors · 2025-08-17
>
> We thank the reviewer for all the thoughtful and constructive comments. They are highly valuable for improving the clarity and scope of our work. Below, we provide point-by-point responses to address each concern and clarify the contributions and limitations of our paper. In addition, we will update our manuscript to incorporate all necessary changes.
>
> [w1] Regarding the “a-priori assessments” and “post-hoc analysis”, these refer to the stability analysis module in SALSA-RL, which is applied after training and does not interfere with the training procedure itself.
>
> Our current implementation focuses on actor-critic methods (SAC and TD3 for the four existing examples, and PPO for the additional Humanoid experiment during rebuttal), but we believe SALSA-RL is algorithm-agnostic in principle, as the action network in DRL architectures can generally be replaced with the proposed latent linear structure. We will clarify these points more explicitly in the revised paper.
>
> [w2] Similar to [w1], SALSA-RL is a modular framework that can be applied on top of standard RL policies and, to the best of our knowledge, can also be extended to other settings such as vision-language-action models. The key is replacing the policy’s action network with a latent linear dynamical system. As long as the original policy can be represented through this latent action structure, the stability analysis remains applicable.
>
> [w3] We thank the reviewer for the insightful comment. In our experiments, we sampled actions either from trained DRL agents or uniformly over the full action space for autoencoder training. However, we have not yet systematically studied the effect of using expert policies from different DRL algorithms.
>
> To further investigate, in this revision, we conducted additional experiments on the Isaac Gym Humanoid task. Even when the autoencoder was trained solely on random action data, SALSA-RL achieved excellent performance, remaining comparable to PPO. This suggests that the method’s generalizability is robust to the choice of pretraining data.
>
> [w4] We thank the reviewer for suggesting evaluation on more complex, high-dimensional control tasks. We agree this is important to demonstrate robustness.
>
> We extended our experiments to Isaac Gym Humanoid (21 actions, 108 observations). We obtained promising results: PPO achieved ~10,400 reward, while SALSA-RL achieved ~9,400, showing comparable performance.
>
> We also emphasize that our autoencoder was trained purely on uniformly sampled grid data (e.g., actions), without relying on data sampled from a trained RL policy. This further demonstrates the generalizability of SALSA-RL.
>
> Consequently, we conducted stability analysis and found that the policy exhibits persistent transitions between stability and instability in action dynamics. Related results are provided in the appendix, along with a supplementary video.
>
> [w5] We thank the reviewer for this valuable comment. Indeed, SALSA-RL currently focuses on post-hoc stability analysis and does not guarantee safety or actively correct instability during training. A natural next step is to extend the framework to inform or guide safety-enhancing actions, potentially in conjunction with safe RL methods. We have incorporated this point into the conclusion, highlighting it as an important direction for future work.

---

> ### Comment · Reviewer_6vq5 · 2025-08-22
>
> I thank the author for making the revision to deal with the concerns. Could you provide the pointer where you made changes in the paper (# of lines or by changing the color of characters)?

---

> > ### Author Response · Authors · 2025-08-22
> >
> > We sincerely thank the reviewer for the suggestion. The manuscript has been revised accordingly, and all major changes have been highlighted in red for clarity.

---

> ### Comment · Reviewer_6vq5 · 2025-08-30
>
> Thank you for the updated manuscripts. The results of Humanoid in IsaacGym seem to be promising.
>
> My remaining concern is that the weak connections to the prior works of safety in deep RL ([w5] in my initial review). In Introduction, discussion, conclusion, related works, etc, I strongly recommend you to emphasize more
> - Why we need to observe the stability in your proposed way?
> - Why the proposal alone (without the discussion about how to control it, or fix it if the system is unstable) is valuable as a publication?
> - Why the recent safe RL is insufficient?

---

> > ### Author Response · Authors · 2025-09-02
> >
> > **Why we need to observe the stability in your proposed way?**
> >
> > We thank the reviewer again for raising this important point. Our method focuses on stability in latent action dynamics, rather than state trajectories:
> > 1. Actions drive state evolution. Inspecting how actions evolve (not just states) can help find oscillations, abrupt reactive patterns, and near-critical behavior.
> > 2.  Action space is often simpler and easier to model.
> > Unlike observation space, which can often be high-dimensional, task-specific, and unknown, the action space (as well as its latent representation) is typically low-dimensional, known, and grounded in actuator commands. In our work, we have used simple autoencoder architectures and benchmark problems to identify this latent space, but significant compression and acceleration in assessing stability can be obtained in real applications with more expressive neural architectures.
> > 3. Actions reveal the policy’s control logic, not just system behavior.
> > State trajectories show what happened, but not why the agent made those decisions. By modeling the action dynamics, we can see the internal regulations and decision patterns of the policy.
> > 4. Actions identify subtle and high-risk behaviors, and provide early warning of failure, as shown in our modified Lunar Lander experiment (Spikes in latent-action instability emerged several steps before any visible state deviation and reliably predicted impending crashes, even though the thrust values themselves remained within their clipped limits).
> > 5. In complex tasks like Humanoid, we capture near-critical operations around 1. These dynamics are not revealed by the standard method but are crucial for understanding robustness.
> >
> > **Why the proposal alone is valuable as a publication?**
> >
> > We appreciate the reviewer’s concern and respectfully emphasize that the core contribution of this work is interpretability and empirical analysis, not control. SALSA-RL is valuable even without intervention capabilities for the following reasons:
> > 1. We provide a post-hoc, non-invasive diagnostic tool that provides a valuable capability in DRL interpretability.
> > SALSA-RL enables empirical stability analysis after training, without requiring any changes to the training loop, reward function, or optimization procedure. This makes it practical for auditing pretrained agents in settings where retraining is costly or infeasible. While standard DRL focuses on performance, it lacks tools to analyze latent instabilities in control behavior. SALSA-RL fills this gap by providing interpretability into the policy’s dynamical structure and deployment readiness, while maintaining performance.
> >
> > 2. SALSA-RL is generalizable and algorithm-agnostic.
> > The method is compatible with a wide range of DRL algorithms, including SAC, DDPG, TD3, and PPO. Besides, we demonstrate its applicability across diverse continuous control environments, from classic control to high-dimensional tasks like Humanoid, making it a practical tool for audit and analysis.
> >
> > 3. It allows for stability analysis without direct interaction. The latent dynamics matrix $A_t = N_\theta(s_t)$ is state-dependent and can be queried offline, without requiring a full simulation for every point. This allows for safety-critical analysis before deployment, such as visualizing regions of instability in the state space, predicting failure zones without interaction, and informing test design and policy selection.
> >
> > **Why the recent safe-RL is insufficient?**
> >
> > We appreciate the reviewer’s comment. The paper presents SALSA-RL not as a replacement for safe RL methods, but as a complementary tool that addresses a different, often overlooked, aspect of safety (post-hoc interpretability and stability analysis).
> >
> > Based on the authors’ understanding, safe RL typically focuses on enforcing safety constraints or correcting instability during training. In contrast, SALSA-RL's primary goal is to provide diagnostic tools to interpret a policy's behavior after training. The goal is not to suggest that safe RL is insufficient, but rather that its focus is different.
> > While safe RL algorithms can produce stable policies, they don't always offer a clear way to analyze why a particular policy is stable or where its potential failure points might be. SALSA-RL directly addresses this by enabling local stability analysis that can identify regions of smooth, risky, or oscillatory behavior in the action-state space. Furthermore, as mentioned in the last point of the above section, SALSA-RL allows exploratory analysis of stability in state-space via the state-dependent linear model formulation. This can inform the operation of the agent far in advance of actual exploration, which carries risks.
> >
> > However, we acknowledge that SALSA-RL currently focuses only on analysis and integration with active safety mechanisms as important future work. We have briefly discussed its connection to safe RL in the conclusion and will elaborate further on this.

---

### Decision · Action_Editor_nyDt · 2025-10-18

**Recommendation:** Reject

**Additional Comments:**

Make a better connection to the prior works in deep RL. The paper proposes how to analyze the stability for deep RL -- currently there is not enough discussion why this is important, and how we can leverage it to improve the safety & stability.

**Audience:**

No

**Audience Explanation:**

The work could be of interest to the Safe RL community if it was better connected to the prior work.

**Claims And Evidence:**

No

**Claims Explanation:**

The paper proposes a novel framework for deep reinforcement learning that models control actions as dynamic, time-dependent variables evolving within a latent space using a pre-trained encoder-decoder and a state-dependent linear system. This design enables non-invasive local stability analysis of pre-trained agents by examining the spectral properties of the latent dynamics matrix, providing policy-level interpretability and identifying regions prone to high-risk, discontinuous behavior.

The reviewers were not completely convinced that the contribution is of sufficient interest to the Safe RL community. Despite authors emphasizing that SALSA-RL is a valuable post-hoc diagnostic tool, critics maintained that a proposal focused purely on stability analysis without demonstrating how to actively control or fix instability was insufficient, and they did not find the concept of latent action stability compelling enough compared to traditional state stability

**Resubmission Of Major Revision:**

The authors may consider submitting a major revision at a later time.